# Interpretable and tractable models of transcriptional noise for the rational design of single-molecule quantification experiments

Gennady Gorin [1,5], John J. Vastola [2,5], Meichen Fang [3] & Lior Pachter [3,4] ✉

The question of how cell-to-cell differences in transcription rate affect RNA count distributions is fundamental for understanding biological processes underlying transcription. Answering this question requires quantitative models that are both interpretable (describing concrete biophysical phenomena) and tractable (amenable to mathematical analysis). This enables the identification of experiments which best discriminate between competing hypotheses. As a proof of principle, we introduce a simple but flexible class of models involving a continuous stochastic transcription rate driving a discrete RNA transcription and splicing process, and compare and contrast two biologically plausible hypotheses about transcription rate variation. One assumes variation is due to DNA experiencing mechanical strain, while the other assumes it is due to regulator number fluctuations. We introduce a framework for numerically and analytically studying such models, and apply Bayesian model selection to identify candidate genes that show signatures of each model in single-cell transcriptomic data from mouse glutamatergic neurons.

Single-cell RNA counts fluctuate due to a combination of dynamic processes in living cells, such as DNA supercoiling, gene regulation, and RNA processing; however, it is unclear how much we can learn about these processes' kinetics and relative importance from counts alone. By generating enormous amounts of single-cell data, modern transcriptomics has the potential to shed light on such fundamental aspects of transcription on a genome-wide scale. However, the field's standard data-driven and phenomenological analyses are descriptive: even though they can summarize data, they do not make specific claims about the mechanisms that generated it. To make mechanistic sense of measurements of gene expression and submolecular features in thousands of single cells at a time[1–4], we seek a framework for systematically distinguishing different plausible hypotheses about transcription.

In principle, models of transcription that are both interpretable and tractable would allow us to be more hypothesis-driven. Interpretability means fitting model parameters conveys clear biological information about the kinetics of microscopic phenomena. Tractability means a thorough mathematical analysis of model behavior is possible. These properties enable a 'rational' design of transcriptomic experiments (Fig. 1a), analogous to ideas about rational drug design[5–9] and the optimal design of single-cell experiments[10–13], since one can mathematically determine the kind of experiment that best distinguishes two such models. One can then perform this experiment, use the results to refine or reject those models, and iterate.

The common post-hoc approach of fitting negative binomial-like distributions to RNA count data[14–18] is mathematically tractable, but not biologically interpretable. On the other hand, detailed mathematical models of transcription[19–27] are certainly interpretable, but tend not to be tractable: complexity makes a thorough analysis challenging, and identifiability issues mean that it can be difficult or impossible to use the data one has to distinguish competing hypotheses.

In this paper, we propose a class of interpretable and tractable transcription models that is fairly simple, yet flexible enough to

[1]Division of Chemistry and Chemical Engineering, California Institute of Technology, Pasadena, CA 91125, USA. [2]Department of Neurobiology, Harvard Medical School, Boston, MA 02115, USA. [3]Division of Biology and Biological Engineering, California Institute of Technology, Pasadena, CA 91125, USA. [4]Department of Computing and Mathematical Sciences, California Institute of Technology, Pasadena, CA 91125, USA. [5]These authors contributed equally: Gennady Gorin and John J. Vastola. ✉e-mail: lpachter@caltech.edu

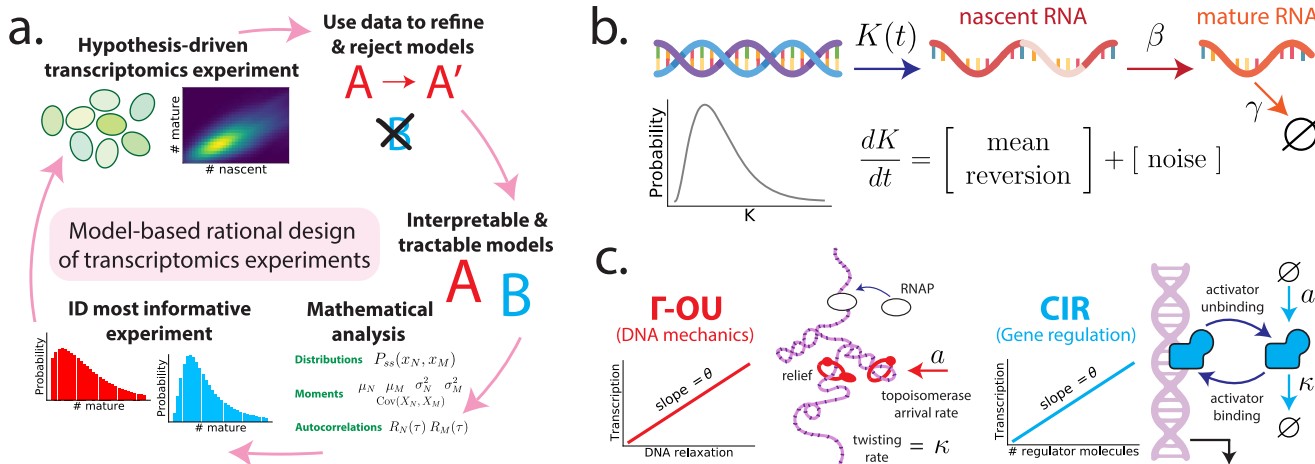

**Fig. 1 | Framework for the rational design of transcriptomics experiments.**
**a** Model-based closed loop paradigm. A researcher begins by representing two or more competing hypotheses as interpretable and tractable mathematical models (middle right of circle). Next, they perform a detailed mathematical analysis of each model, computing quantities (e.g., RNA count distributions and moments) that can help distinguish one hypothesis from another. Using the results of that analysis as input, they identify the experiment that best distinguishes the two models. Finally, they perform this experiment on some population of cells, use the resulting data to refine and/or reject models, and repeat the process with an updated ensemble of models. **b** Interpretable and tractable modeling framework for transcription rate variation. We consider stochastic models of transcription involving (i) nascent/unspliced RNA, (ii) mature/spliced RNA, and (iii) a stochastic

and time-varying transcription rate $K(t)$. The transcription rate is assumed to evolve in time according to a simple, one-dimensional SDE that includes a mean-reversion term (which tends to push $K(t)$ towards its mean value) and a noise term (which causes $K(t)$ to randomly fluctuate). Here, we have specifically chosen dynamics for which the long-time probability distribution of $K(t)$ is a gamma distribution (gray curve), because this assumption yields empirically plausible negative binomial-like RNA distributions. However, the framework does not require this in general. **c** Two plausible models studied in this paper. The gamma Ornstein–Uhlenbeck (Γ-OU) model describes DNA mechanics, whereas the Cox–Ingersoll–Ross (CIR) model describes regulation by a high copy number regulator.

account for a range of biological phenomena. It assumes that a stochastic and time-varying transcription rate drives a discrete stochastic RNA transcription and splicing process. This model class incorporates both intrinsic noise (randomness associated with the timing of events like transcription and degradation) and extrinsic noise (due to cell-to-cell differences)[28–32] in a principled way, with the latter due to transcription rate variation. We focus on two specific examples of models from this class, which assume variation is due to (i) random changes in the mechanical state of DNA, or (ii) random changes in the number of an abundant regulator.

We find that these models, although mathematically similar, yield different predictions; this indicates that the fine details of transcription can, at least some of the time, be inferred from transcription rate variation. This is because the details of how the transcription rate fluctuates (i.e., its dynamics), rather than just the steady-state distribution of those fluctuations, can qualitatively affect model predictions. We also find that a naïve moment-based approach to distinguishing between them fails, and that comparing whole joint distributions far outperforms other approaches. Finally, as a proof of principle, we fit these models to single-cell transcriptomic data from mouse glutamatergic neurons, and find in many cases that (i) the two models can be distinguished, and that (ii) they greatly outperform standard negative binomial-like fits, even when one penalizes model complexity using a principled Bayesian model selection procedure. While we will not actually implement the entire closed loop paradigm depicted in Fig. 1a, our work constructs one possible mathematical and computational foundation for it.

## Results

### Transcription rate variation accounts for empirically observed variance
If we would like to understand and fit available transcriptomic data–especially multimodal data sets that report the numbers of both nascent and mature transcripts inside single cells[33,34]–what kind of models of transcription should we consider? Given that single cell RNA counts

are often low, we would like our models to be able to account for the production, processing, and degradation of individual RNA molecules. From experiments in living cells, these processes are known to be random[35]. Crucially, the molecule counts are low enough that the variation in molecule numbers should be explicitly described by a stochastic model[36].

The theoretical framework associated with the chemical master equation (CME)[37–43] can be used to define discrete and stochastic models of cellular processes. The *constitutive* model of transcription, which assumes RNA is produced at a constant rate, is one particularly simple and well-studied example. It can be defined via the chemical reactions

$$\varnothing \xrightarrow{K} \mathcal{N} \xrightarrow{\beta} \mathcal{M} \xrightarrow{\gamma} \varnothing, \tag{1}$$

where $\mathcal{N}$ denotes nascent RNA, $\mathcal{M}$ denotes mature RNA, $K$ is the transcription rate, $\beta$ is the splicing rate, and $\gamma$ is the degradation rate. It predicts[44,45] that the long-time probability $P_{ss}^{con}(x_N, x_M)$ of observing $x_N \in \mathbb{N}_0$ nascent RNA and $x_M \in \mathbb{N}_0$ mature RNA in a single cell is Poisson, so that

$$P_{ss}^{con}(x_N, x_M) = \frac{\left(\frac{K}{\beta}\right)^{x_N} e^{-K/\beta}}{x_N!} \frac{\left(\frac{K}{\gamma}\right)^{x_M} e^{-K/\gamma}}{x_M!}. \tag{2}$$

While mathematically tractable, a model like this is too simple to fit existing data. Most observed eukaryotic RNA count distributions are 'overdispersed': they have a higher variance than Poisson distributions with the same mean[46].

One way to account for overdispersion is to assume that different cells in a population have different transcription rates, but that each individual cell otherwise follows the constitutive model. For various choices of transcription rate distribution, one can obtain results that look much closer to eukaryotic transcriptomic data. For example, one reasonable choice (which has been explored by other authors[47]) is to

assume that the transcription rate $K$ is gamma-distributed with shape parameter $\alpha$ and scale parameter $\theta$, i.e., $K \sim \Gamma(\alpha, \theta)$. The long-time/steady-state probability of observing $x_N$ nascent and $x_M$ mature RNA would then be described by the Poisson-gamma *mixture* model

$$P_{ss}^{mix}(x_N, x_M) = \int_0^\infty dK \, \frac{K^{\alpha-1} e^{-K/\theta}}{\theta^\alpha \, \Gamma(\alpha)} P_{ss}^{con}(x_N, x_M). \qquad (3)$$

The marginal distributions of this joint distribution will be negative binomial rather than Poisson, allowing us to actually fit observed single-cell data. But this approach—which is equivalent to the post-hoc fitting of negative binomial distributions—is not biophysically interpretable. What is the biological meaning of the parameters $\alpha$ and $\theta$? And why do different cells have different transcription rates? Is it really reasonable to assume, as we have here, that these rates are 'frozen', and remain as they are for all time in a given cell?

## Interpretable and tractable modeling framework for transcription rate variation

We propose a class of transcriptional models that balance interpretability and tractability, and generalize the mixture model. Although various biological details underlying transcription may be complicated, we assume they can be captured by an effective transcription rate $K(t)$ which is stochastic and varies with time. This transcription rate randomly fluctuates about its mean value, with the precise nature of its fluctuations dependent upon the fine biophysical details of transcription. Mathematically, we assume that $K(t)$ is a continuous-valued stochastic process described by an (Itô-interpreted) stochastic differential equation (SDE)

$$\dot{K}(t) = [\text{mean reversion}] + [\text{noise}]$$
$$= A - BK(t) + [\text{noise}] \qquad (4)$$

for some coefficients $A$ and $B$, where [mean reversion] denotes a deterministic term that drives the transcription rate towards its mean value, and [noise] denotes a model-dependent term that introduces stochastic variation. The transcription rate $K(t)$ is coupled to RNA dynamics as in the constitutive model:

$$\varnothing \xrightarrow{K(t)} \mathcal{N} \xrightarrow{\beta} \mathcal{M} \xrightarrow{\gamma} \varnothing. \qquad (5)$$

This reaction list defines a master equation model that couples discrete stochastic RNA dynamics to the continuous stochastic process $K(t)$ (Fig. 1b). Although this model class is not completely realistic (for example, there is no feedback), it is fairly flexible, and can recapitulate empirically plausible negative binomial-like RNA count distributions. To guarantee this, we will specifically consider candidate models for which the steady-state distribution of $K(t)$ is a gamma distribution.

Other kinds of transcriptional models can also be viewed as special cases of this model class. The constitutive model (Eq. (1)) is a degenerate case that arises from the limit of no noise and fast mean-reversion, and the mixture model arises from the limit of slow transcription rate variation. We will see later that the popular bursting model of RNA production, which describes intermittent production of multiple nascent transcripts at a time[1,48–51] is also a degenerate case. For the rest of this paper, we examine two specific cases of this model class more closely: the gamma Ornstein-Uhlenbeck (Γ-OU) model and Cox-Ingersoll-Ross (CIR) model, which are depicted in Fig. 1c. In particular, we will motivate the underlying biophysics, solve the models, outline major similarities and differences, and discuss how and when they can be distinguished given transcriptomic data.

Coupling upstream variability to transcriptional CMEs has been studied before (e.g., by Dattani and Barahona[52]), but usually in a way

that assumes either that $K(t)$ takes on a finite set of values (for example, gene switching[53]), or that the distribution of $K(t)$ is a priori known, rather than defined by a stochastic dynamical system like Eq. (4). We attempt to build on these studies by treating $K(t)$ as a continuous stochastic dynamical variable on the same footing as nascent and mature RNA counts.

### A. Gamma Ornstein–Uhlenbeck production rate model

Transcription rate variation may emerge due to mechanical changes in DNA that make producing RNA more or less kinetically favorable. Each nascent RNA produced by an RNA polymerase induces a small amount of mechanical stress/supercoiling in DNA, which builds over time and can mechanically frustrate transcription unless it is relieved. Because topoisomerases arrive to relieve stress (Fig. 1c), there is a dynamic balance between transcription-mediated stress and topoisomerase-mediated recovery, models of which can recapitulate gene over-dispersion and bursting[23,25].

We can simplify the detailed mechanistic model of Sevier, Kessler, and Levine while retaining crucial qualitative aspects. For example, we can model transcriptional catalysis at a promoter $\mathcal{G}$ by a reservoir of RNA polymerase $\mathcal{P}$:

$$\mathcal{P} + \mathcal{G} \xrightarrow{k_{ini}} \mathcal{P} + \mathcal{G} + \mathcal{N}, \qquad (6)$$

where $k_{ini}$ is the rate of transcription initiation. Next, we assume that $k_{ini}$ is proportional to the DNA relaxation state: if the DNA is in a stressed, twisted state, polymerase binding events are less likely to succeed. We propose that relaxation continuously decreases due to transcription-associated events, and that topoisomerases randomly arrive to increase relaxation according to an exponential law. The direct proportionality between the amount of DNA relaxation and $k_{ini}$ is a coarse, first-order approximation valid when $k_{ini}$ is small. This approximation may be biophysically justified by appealing to the prevalence of DNA compaction in eukaryotic cells. If the concentration $p$ of RNA polymerase is high and its variation is low, we find (see Section 3.2.1 in the Supplementary Note) that the overall transcription rate $K(t)$ can be modeled by the SDE

$$\dot{K}(t) = -\kappa K(t) + \epsilon(t; a, \theta), \qquad (7)$$

where $\epsilon(t; a, \theta)$ is an infinitesimal Lévy process (a compound Poisson process with arrival frequency $a$ and exponentially distributed jumps with expected size $\theta$) capturing random topoisomerase arrival. This is the gamma Ornstein–Uhlenbeck (Γ-OU) model of transcription[54]. It naturally emerges from a biomechanical model with two opposing effects: the continuous mechanical frustration of DNA undergoing transcription, which is a first-order process with relaxation rate $\kappa$, and the stochastic relaxation by topoisomerases that arrive at rate $a$. The scaling between the relaxation state and the transcription rate is set by a gain parameter $\theta \propto \langle p \rangle$, where $\langle p \rangle$ is average polymerase concentration; its coefficient of proportionality includes the coefficient of the aforementioned first-order expansion of $k_{ini}$ as a function of the amount of DNA relaxation.

The Γ-OU model is perhaps better known in finance applications, where it has been used to model the stochastic volatility of the prices of stocks and options[55–58]. Its utility as a financial model is largely due to its ability to capture asset behavior that deviates from that of commonly used Gaussian Ornstein–Uhlenbeck models, such as skewness and frequent price jumps.

### B. Cox–Ingersoll–Ross production rate model

Alternatively, transcription rate variation may be due to non-negligible fluctuations in the concentration of a regulator $\mathcal{R}$. We can encode

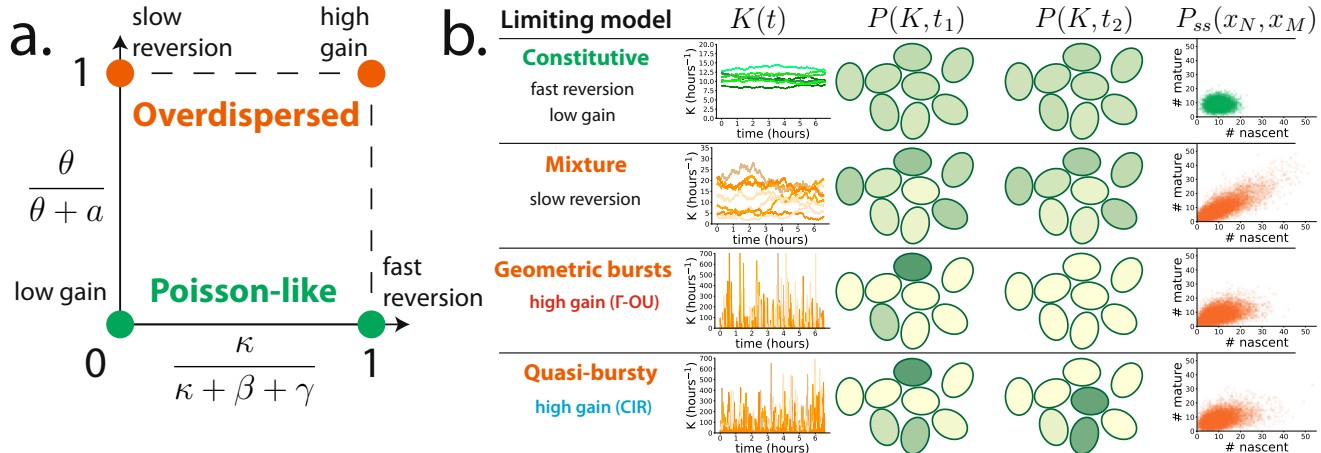

**Fig. 2 | Summary of the qualitative behavior of the Γ-OU and CIR models.**
**a** Qualitative behavior can be visualized in a two-dimensional parameter space, with $\kappa/(\kappa + \beta + \gamma)$ on one axis and the gain ratio $\theta/(\theta + a)$ on the other. The four limits discussed in the text correspond to the four corners of this space. When $a \gg \theta$, we obtain Poisson-like behavior (green). When $a \ll \theta$, we obtain overdispersed distributions (orange). **b** Dynamics of limiting models. The Γ-OU and CIR models were simulated using four parameter sets close to the limiting regimes; transcription rates are visualized using trajectories and cell cartoons, where transcription rate is a logarithmic function of cell color. Ten thousand samples from the joint RNA

count distribution are depicted in the rightmost column. Both models reduce to the constitutive model in the fast reversion and low gain limits, where the transcription rate $K(t)$ is effectively constant in time and identical for all cells in the population. Both reduce to the mixture model in the slow reversion limit, so that $K(t)$ is inhomogeneous across the population but constant in time for individual cells. In the high gain limit, the Γ-OU and CIR models yield different heavy-tailed distributions, with the CIR limiting model appearing to be uncharacterized. In both cases, $K(t)$ exhibits sporadic large fluctuations within single cells.

these fluctuations by defining a multi-state promoter $\mathcal{G}$ activated by $\mathcal{R}$:

$$\varnothing \xrightarrow{a} \mathcal{R} \xrightarrow{\kappa} \varnothing$$

$$\mathcal{G}_{off} + \mathcal{R} \underset{k_{off}}{\overset{k_{on}}{\rightleftharpoons}} \mathcal{G}_{on} \qquad (8)$$

$$\mathcal{G}_{on} \xrightarrow{k_{ini}} \mathcal{G}_{on} + \mathcal{N},$$

where $a$ is the $\mathcal{R}$ production rate and $\kappa$ is the $\mathcal{R}$ degradation rate. If the number of regulator molecules $r(t)$ is very large, we can accurately approximate regulator birth and death dynamics as a real-valued stochastic process using the framework associated with the chemical Langevin equation (CLE)[39,59]. Under the assumptions of rapid, weak binding, the effective transcription rate $K(t) := \theta r(t)$ satisfies the SDE

$$\dot{K}(t) = a\theta - \kappa K(t) + \sqrt{2\kappa\theta K(t)}\,\xi(t) \qquad (9)$$

where $\xi(t)$ is a Gaussian white noise term and $\theta = k_{ini}k_{on}/k_{off}$ (see Section 3.3.1 in the Supplementary Note). This is the Cox–Ingersoll–Ross (CIR) model of transcription[54].

Although the CIR model is most familiar as a description of interest rates in quantitative finance[60–62], it has been previously used to describe biochemical input variation based on the CLE, albeit with less discussion of the theoretical basis and limits of applicability[63–66].

### The models are interpretable and unify known results

Qualitatively, the distribution shapes predicted by the Γ-OU and CIR models interpolate between Poisson and negative binomial-like extremes, with behavior controlled mostly by two of the transcription noise parameters: the mean-reversion rate $\kappa$ and the gain parameter $\theta$ (Fig. 2a). Remarkably, where one is in this landscape of qualitative behavior is independent of the mean transcription rate $\langle K \rangle = a\theta/\kappa$, since $a$ can vary to accommodate any changes in $\kappa$ or $\theta$. It is also independent of the steady-state distribution of transcription rates, which is the same (i.e., $\Gamma(a/\kappa, \theta)$) in all cases. We find that the details of how the transcription rate fluctuates in time strongly impact the shape of RNA count distributions, a fact which may have previously gone underappreciated.

When $\kappa$ is very fast, the transcription rate very quickly reverts to its mean value whenever it is perturbed, so it is effectively constant, and we recover the constitutive model. When $\kappa$ is very slow, the transcription rates of individual cells appear 'frozen' on the time scales of RNA dynamics, and we recover the mixture model discussed earlier. When $\theta$ is very small, fluctuations in underlying biological factors (DNA relaxation state or regulator concentration) are significantly damped, so $K(t)$ is also effectively constant in this case.

Interestingly, while the two models agree in the aforementioned limits, their predictions markedly differ in the large $\theta$ limit, where fluctuations are amplified and predicted count distributions become increasingly overdispersed. The Γ-OU model predicts that nascent RNA is produced in geometrically distributed bursts in this limit, recapitulating the conventional model of bursty gene expression[35,51]. However, the CIR model predicts a previously uncharacterized family of count distributions with heavier tails than their Γ-OU counterparts. The difference is shown in Supplementary Fig. 4. This deviation is a consequence of state-dependent noise: while the number of topoisomerases which arrive to relieve stress does not depend on the current relaxation state of the DNA, birth-death fluctuations in the number of regulators tend to be greater when there are more regulator molecules present. We illustrate the four limiting regimes of interest in Fig. 2b, present their precise quantitative forms in Section 2.5, and derive them in Section 5 in the Supplementary Note.

Another lens through which to view qualitative behavior is the squared coefficient of variation ($\eta^2 := \sigma^2/\mu^2$), which quantifies the amount of 'noise' in a system. We derived the exact result that (see Section 2.4.2 in the Supplementary Note), consistently with previous studies[29,30,32], the total noise can be written as a sum of 'intrinsic' (due to the stochasticity inherent in chemical reactions) and 'extrinsic' (due to transcription rate variation) contributions. For both models,

$$\eta_N^2 = \frac{1}{\mu_N} + \frac{\theta}{\langle K \rangle}\frac{1/\kappa}{1/\kappa + 1/\beta}$$

$$\eta_M^2 = \frac{1}{\mu_M} + \frac{\theta}{\langle K \rangle}\frac{1/\kappa}{1/\kappa + 1/\beta}\frac{1/\kappa}{1/\kappa + 1/\gamma}\frac{1/\kappa + 1/(\beta+\gamma)}{1/\kappa}, \qquad (10)$$

where $\eta_N^2$ and $\eta_M^2$ quantify the amount of noise in nascent and mature RNA counts, and $\mu_N$ and $\mu_M$ denote the average number of nascent and mature RNA. In the 'overdispersed' regimes, where $\theta$ is large or $\kappa$ is small, the extrinsic noise contributions become significant, but not in a way that maps cleanly onto the space depicted in Fig. 2a. For example, the fraction of extrinsic noise for nascent RNA is

$$(\text{extrinsic fraction})_N := \frac{\eta_N^2 - \frac{1}{\mu_N}}{\eta_N^2} = \frac{\theta}{\theta + \kappa + \beta} \qquad (11)$$

whose relative size in different overdispersed regimes changes depending on the splicing rate $\beta$. The behavior of the extrinsic noise fraction as a function of the parameters is summarized in Supplementary Fig. 5.

### The models are analytically tractable

Using a suite of diverse theoretical approaches—including path integral methods, generating function computations, a correspondence between the Poisson representation of the CME and SDEs, and tools from the mathematics of stochastic processes—we were able to exactly solve the Γ-OU and CIR models. This includes computing all steady-state probability distributions $P_{ss}(x_N, x_M)$, first-order moments, second-order moments, and autocorrelation functions.

A central idea in all of our calculations is to consider transforms of the probability distribution—variants of the generating function—instead of the distribution itself. Once a generating function is available, the distribution can be obtained by computationally inexpensive Fourier inversion. The joint generating function $\psi(g_N, g_M, h, t)$ is defined as

$$\psi := \sum_{x_N=0}^{\infty} \sum_{x_M=0}^{\infty} \int_0^{\infty} dK\, g_N^{x_N} g_M^{x_M} e^{ihK}\, P(x_N, x_M, K, t), \qquad (12)$$

with $g_N, g_M \in \mathbb{C}$ both on the complex unit circle, $h \in \mathbb{R}$, and $P(x_N, x_M, K, t)$ encoding the probability density over counts and transcription rates. As these rates are not usually observable, and the previous body of work treats stationary distributions, we are most interested in $\psi_{ss}(g_N, g_M)$, the probability-generating function (PGF) of $P_{ss}(x_N, x_M)$. We find it most convenient to report our results in terms of $\phi_{ss}(u_N, u_M) := \log \psi_{ss}(g_N, g_M)$, the log of the PGF with an argument shift $u_N := g_N - 1$ and $u_M := g_M - 1$.

The solution of the Γ-OU model is

$$\phi_{ss}(u_N, u_M) = \langle K \rangle \int_0^{\infty} \frac{U_0(s; u_N, u_M)}{1 - \frac{\theta}{\kappa} U_0(s; u_N, u_M)}\, ds, \qquad (13)$$

where $U_0(s; u_N, u_M)$ is obtained by solving the characteristic ODEs obtained from the generating function[67]:

$$\begin{aligned}
\frac{dU_2}{ds} &= -\gamma U_2, & U_2(0) &= u_M, \\
\frac{dU_1}{ds} &= \beta(U_2 - U_1), & U_1(0) &= u_N, \\
\frac{dU_0}{ds} &= \kappa(U_1 - U_0), & U_0(0) &= 0.
\end{aligned} \qquad (14)$$

This system of linear first-order ODEs can be solved analytically[50], and the generating function can be obtained by quadrature. The solution to the CIR model is

$$\phi_{ss}(u_N, u_M) = \langle K \rangle \int_0^{\infty} U_0(s; u_N, u_M)\, ds, \qquad (15)$$

**Table 1 | Molecular distribution moments**

| Moment | Value |
|---|---|
| $\langle K \rangle$ | $\frac{a\theta}{\kappa}$ |
| $\mu_N$ | $\langle K \rangle / \beta$ |
| $\mu_M$ | $\langle K \rangle / \gamma$ |
| $\sigma_N^2 - \mu_N$ | $\frac{\mu_N \theta}{\kappa + \beta}$ |
| $\sigma_M^2 - \mu_M$ | $\frac{\mu_M \theta}{\kappa + \gamma} \cdot \frac{\beta}{\kappa + \beta} \cdot \frac{\kappa + \beta + \gamma}{\beta + \gamma}$ |
| $\text{Cov}(X_N, K)$ | $\frac{\langle K \rangle \theta}{\kappa + \beta}$ |
| $\text{Cov}(X_M, K)$ | $\frac{\langle K \rangle \theta}{\kappa + \gamma} \cdot \frac{\beta}{\kappa + \beta}$ |
| $\text{Cov}(X_N, X_M)$ | $\frac{\langle K \rangle \theta}{(\kappa + \beta)(\kappa + \gamma)} \cdot \frac{\kappa + \beta + \gamma}{\beta + \gamma}$ |
| $R_N(\tau)$ | $e^{-\beta\tau} + \frac{\text{Cov}(X_N, K)}{\sigma_N^2} \left[ \frac{e^{-\kappa\tau} - e^{-\beta\tau}}{\beta - \kappa} \right]$ |
| $R_M(\tau)$ | $e^{-\gamma\tau} + \beta \frac{\text{Cov}(X_N, X_M)}{\sigma_M^2} \left[ \frac{e^{-\beta\tau} - e^{-\gamma\tau}}{\gamma - \beta} \right] + \beta \frac{\text{Cov}(X_M, K)}{\sigma_M^2}$ |
| | $\times \left[ \frac{e^{-\beta\tau}}{(\beta - \gamma)(\beta - \kappa)} + \frac{e^{-\gamma\tau}}{(\gamma - \beta)(\gamma - \kappa)} + \frac{e^{-\kappa\tau}}{(\kappa - \beta)(\kappa - \gamma)} \right]$ |

where $U_0(s; u_N, u_M)$ is obtained from analogous ODEs:

$$\begin{aligned}
\frac{dU_2}{ds} &= -\gamma U_2, & U_2(0) &= u_M, \\
\frac{dU_1}{ds} &= \beta(U_2 - U_1), & U_1(0) &= u_N, \\
\frac{dU_0}{ds} &= \kappa(U_1 - U_0) + \theta U_0^2, & U_0(0) &= 0.
\end{aligned} \qquad (16)$$

While the above ODEs have an exact solution, it is cumbersome, and preferable to evaluate numerically. We derive these solutions in Section 3, and validate them against stochastic simulations in Section 6 in the Supplementary Note.

### Summary statistics cannot distinguish between the models

The tractability of these two models allows us to analytically compute common (steady-state) summary statistics. Despite the models' distinct biological origins, their means ($\mu_N$ and $\mu_M$), variances ($\sigma_N^2$ and $\sigma_M^2$), covariances, and autocorrelation functions ($R_N(\tau)$ and $R_M(\tau)$) match exactly (Table 1; see Section 4 in the Supplementary Note). This means that such summary statistics cannot be used as the basis for model discrimination. More fundamentally, it implies that experimental technologies that only report averages—such as RNA sequencing without single-cell resolution—cannot possibly distinguish between noise models.

### Models can be distinguished using multimodal count data

Even if our two models did not have identical first and second order moments, the shortcomings of moment-based model discrimination are becoming increasingly clear[68]. Does the situation improve if we compare whole count distributions? To establish that the Γ-OU and CIR models are in principle discriminable, we performed an in silico experiment: first, (i) we generated noise-free synthetic data from the CIR model for many different parameter sets; then (ii) we compared the goodness-of-fit of each model to this synthetic data.

We chose to quantify the relative goodness-of-fit of each model via the *Bayes factor*, which is the ratio of the likelihood of each model given the data. Concretely, we computed the log Bayes factor

$$\log_{10} \text{BF} := \log_{10} \left[ \frac{P(\text{data}|\text{CIR})}{P(\text{data}|\Gamma-\text{OU})} \right] \qquad (17)$$

for each of the synthetic data sets we considered. A log Bayes factor of close to zero means that neither model is preferred, while a log Bayes factor of magnitude at least 2—i.e., one model is at least a hundred times more likely than the other—is commonly considered decisive

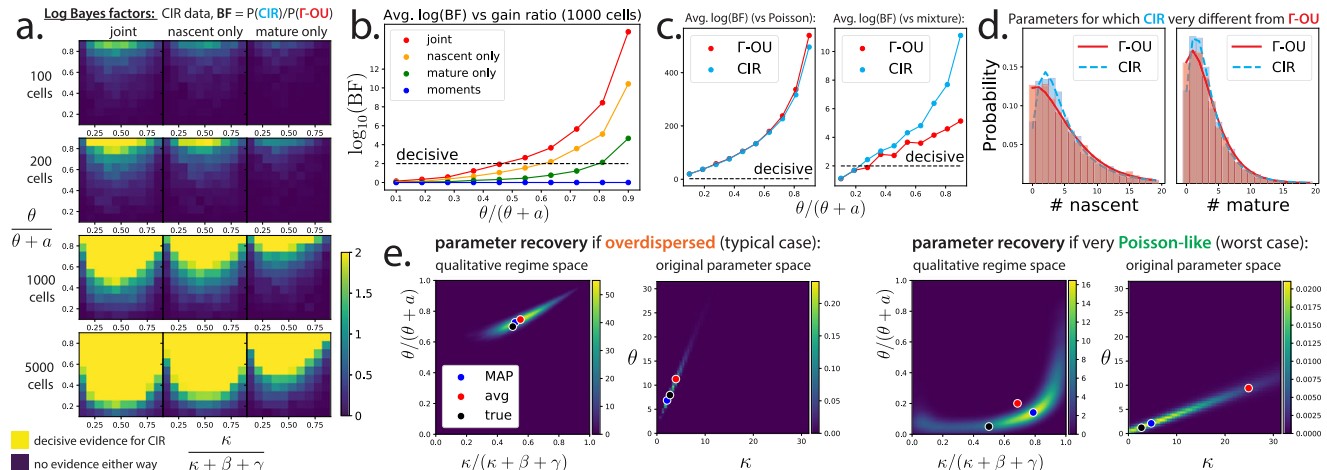

**Fig. 3 | Model distinguishability and parameter inference. a** Log Bayes factors show models are distinguishable in most of parameter space. Plotted are the average log Bayes factors capped at 2 (a common threshold for decisive evidence in favor of one model). **b** Models are often strongly distinguishable. A slice of the 1000 cell row of the previous plot (without the cap at 2) for a moderate value of $\kappa/(\kappa+\beta+\gamma)$. Using both nascent and mature data is better than using either individually, usually by at least an order of magnitude. **c** It is easy to distinguish the Γ-OU and CIR models from trivial models. Same axes as in (**b**). Discriminability of Γ-OU and CIR models versus Poisson and mixture models for a somewhat small value of $\kappa$ ($\kappa/(\kappa+\beta+\gamma) = 0.1$), where discriminability is expected to be difficult. **d** Nascent and mature marginal distributions for the Γ-OU (red) and CIR (blue) models for a maximally distinguishable parameter set. Histograms show synthetic data (5000 cells), while the smooth lines show the exact results. **e** Bayesian inference of noise model parameters. We sampled the posterior distribution of the parameters of the Γ-OU model (assuming it is known that $\beta = 1$, $\gamma = 1.7$, and $\langle K \rangle = 10$), given a synthetic data set of 1000 cells. Posteriors are presented in both the qualitative regimes space, and in terms of the original parameters. For very Poisson-like data, posteriors are broad in both spaces, because $\kappa$ is no longer identifiable. MAP: mode of posterior, avg: average of posterior, true: true parameter values.

evidence that one model is superior, and will be used as our criterion for distinguishability.

Because we expect that model distinguishability primarily depends on distribution 'shape' (e.g., Poisson-like or negative binomial-like), and because shape appears to be controlled by where a parameter set resides in the two-dimensional space depicted in Fig. 2, we chose 100 parameter sets that uniformly cover this space. Parameters which do not strongly control distribution shape (the average transcription rate, splicing rate, and degradation rate) were held fixed.

For each parameter set, we generated 100 synthetic data sets, and averaged the corresponding log Bayes factors over them to account for sampling noise (Fig. 3a). We varied the number of cells per synthetic data set, and computed Bayes factors using (i) full joint distributions, (ii) nascent distributions only, and (iii) mature distributions only. As expected, distinguishability is higher when data sets have more cells, when multimodal data is used, and when the data are overdispersed rather than Poisson-like. Data from ≈1000 cells is required for the models to be distinguishable in most of parameter space. Using multimodal data instead of nascent or mature counts only can improve distinguishability by about an order of magnitude (Fig. 3b) on average. Changing the ground truth model (here, CIR) or values of the parameters held fixed ($\langle K \rangle$, $\beta$, $\gamma$) does not qualitatively change the results (Supplementary Fig. 6).

We would also like each model to be individually discriminable from 'trivial' models; one advantage of using Bayes factors to quantify discriminability here is that they automatically penalize model complexity. Using the same data sets, we compared each model with the constitutive and mixture models, and found generally strong distinguishability (Fig. 3c), with particularly high Bayes factors when comparing against the constitutive model.

How different are the predictions of the Γ-OU and CIR models when they are maximally distinguishable? For the parameter set with the maximum log Bayes factor, we found that the predicted distributions are still visually alike (Fig. 3d). This illustrates that, while probabilistic inference using whole distributions may succeed at performing model discrimination, many naïve approaches, such as those based on moments or single marginals, may fail.

## Accurate parameter recovery is possible

Even if one can distinguish between the two noise models, it is possible that the biophysically interesting parameters controlling transcription rate variation (e.g., $\kappa$ and $\theta$) cannot be precisely inferred from steady-state RNA count data. In fact, ambiguity is fairly expected, since qualitative behavior only strongly depends on certain parameter ratios near limiting regimes (see Fig. 2 and Section 2.5 in the Supplementary Note). For example, when the gain ratio $\theta/(\theta + a)$ is small, each model predicts Poisson-like RNA count distributions, which are not very sensitive to the value of $\kappa$.

To illustrate the conditions under which parameter recovery appears to be possible, we performed an in silico parameter recovery experiment with two parameter sets: one which is overdispersed, as transcriptomic data tends to be, and another which is very Poisson-like. For each parameter set, a noise-free synthetic data set of 1000 cells was generated from the Γ-OU model, and then a Bayesian parameter recovery analysis was performed to construct a posterior distribution of parameters that could have generated the data. One advantage of a Bayesian approach is that, in addition to obtaining a point estimate of the most likely parameter set given the data, one obtains a measure of uncertainty from the spread of this distribution.

We find that, in the typical scenario (overdispersed data), the posterior is fairly tight in both the qualitative regime space and in the original parameter space (Fig. 3e). Both $\kappa$ and $\theta$ are fairly identifiable, allowing us to be optimistic that it is possible to infer biophysical parameters related to transcription rate variation from single-cell data. In the pessimistic scenario (Poisson-like data), model predictions appear to be 'sloppy' with respect to $\kappa$, as expected, yielding a broad distribution of possible $\kappa$ and $\theta$ values.

## Multimodal count distributions in sequencing datasets suggest distinct modes of transcriptional regulation

Even if the Γ-OU and CIR models can be distinguished and fit to data in principle, can they be distinguished and fit in practice? Real transcriptomic data feature additional noise due to technical errors[69], and possibly confounding influences due to phenomena like cell growth and division[70]. One can also face serious model misspecification

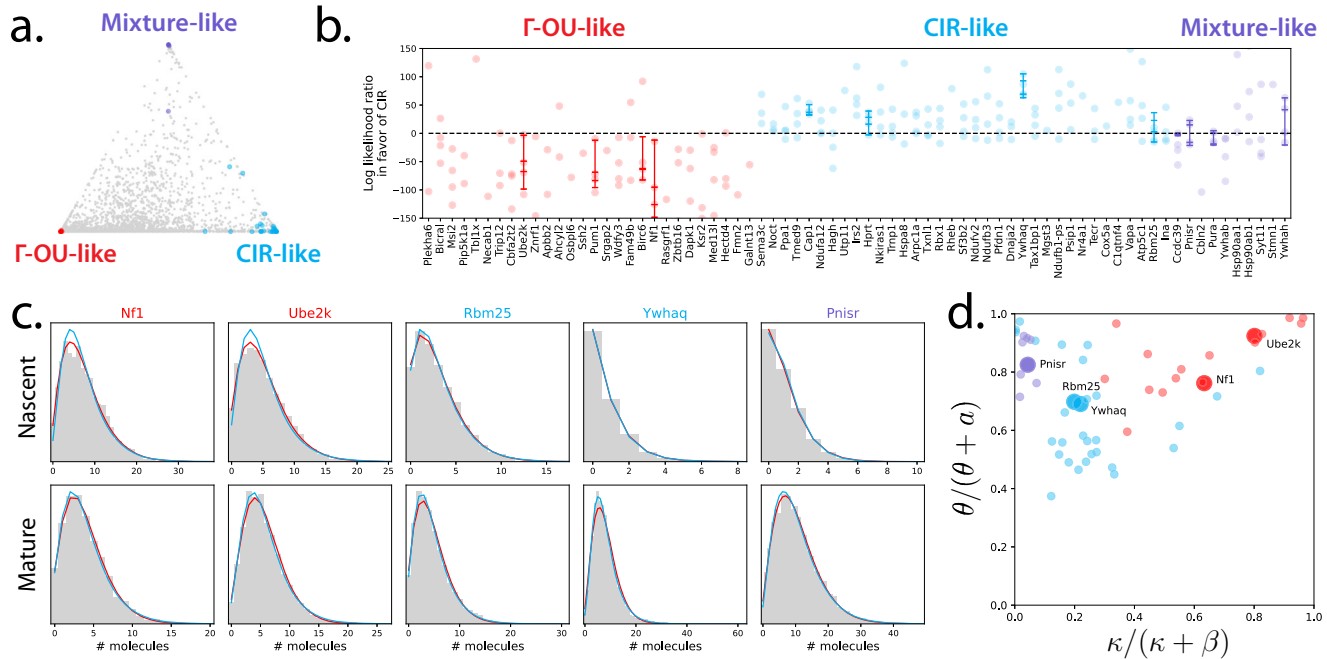

**Fig. 4 | Genes from comparable single-cell RNA sequencing datasets can be consistently assigned to a particular biophysical model of transcription. a** By fitting models in the limiting regimes and calculating model Akaike weights, visualized on a ternary diagram, we can obtain coarse gene model assignments (colors: regimes predicted by the partial fit; red: Γ-OU-like genes; blue: CIR-like genes; violet: mixture-like genes; gray: genes not consistently assigned to a limiting regime). **b** Likelihood ratios for selected genes are consistent across biological replicates, and favor categories consistent with predictions (colors: regimes predicted by the partial fit; points: likelihood ratios; horizontal line markers: Bayes factors; vertical lines: Bayes factor ranges; Bayes factor values beyond the plot bounds have been omitted. $n = 4$ biologically independent animals, with 5343, 6604, 5892, and 4497 cells per animal). **c** The differences between model best fits are reflected in raw count data (title colors: predicted regimes; lines: model fits at maximum likelihood parameter estimates; line colors: models; histograms: count data). **d** Non-distinguishable genes tend to lie in the slow-reversion and high-gain parameter regime; distinguishable genes vary more, but tend to have relatively high gain (colors: predicted regimes, large dots: genes illustrated in panel (**c**). Genes with absolute log-likelihood ratios above 150 have been excluded).

problems, where one finds that even though one model fits better than others, none of them fit particularly well.

To show that these models may be observed and distinguished in real datasets, we analyzed single-cell transcriptomic data with tens of thousands of genes from the glutamatergic neurons of four mice[71] (31,649 genes after pseudoalignment yielding unspliced and spliced RNA counts[72,73]). Because neurons generally do not grow or divide, their gene expression dynamics should not be confounded by the effects of cell growth and division. To guard against spurious conclusions related to both technical noise and model misspecification, we used a multi-step filtering procedure based on neuron subtypes from a single mouse dataset to choose genes to examine ("Methods" and Section 8 in the Supplementary Note).

We fit the Γ-OU and CIR models to the 80 genes that passed the filtering step (Fig. 4a) to data from four mice, using gradient descent to find the maximum likelihood parameter set, and using likelihood ratios for model selection. We discarded all results with absolute log-likelihood ratios above 150, as they appeared to reflect failure to converge to a satisfactory optimum (Supplementary Figs. 8–37). The likelihood ratios for the remaining 73 genes are depicted in Fig. 4b (points). To ensure that the likelihood ratios we obtained were not distorted by the omission of uncertainty in estimates, or potentially suboptimal fits, we further fit twelve of the genes using a Bayesian procedure like the one used in Fig. 3e, demonstrating the distribution of Bayes factors in the same axes (horizontal markers).

The predictions from the coarse filter were largely concordant with the results from the full model, suggesting that it is effective for selecting genes of interest from transcriptome-wide data. The model assignments were typically consistent among datasets. Although orthogonal targeted experiments are necessary to identify whether the proposed models effectively recapitulate the live-cell transcriptional dynamics, the reproducibility of the findings suggests directions and candidate genes for such investigations. Finally, the Bayes factors were largely quantitatively consistent with the likelihood ratios, suggesting that the approximations made in the gradient descent procedure do not substantially degrade the quality of the statistical results. However, we did observe several discrepancies between likelihood ratios and Bayes factors, confirming that the more computationally facile gradient descent procedure does not perfectly recapitulate the full Bayesian fit (cf. results for Ccdc39 and Birc6), possibly due to substantial omitted uncertainty in some genes' parameters.

Five example fits are depicted in Fig. 4c, with the corresponding gene names color-coded according to the best-fit model (red: Γ-OU, blue: CIR, purple: mixture). The results for all genes and datasets are shown in Supplementary Figs. 8–37. Model discrepancies mostly appear to be due to differences in probability near distribution peaks. Interestingly, only either the nascent marginal or mature marginal exhibits obvious visual differences between model fits in some of the genes depicted here, further motivating the use of multimodal data.

The location of each best-fit parameter set in the qualitative regimes space is shown in Fig. 4d. Most Γ-OU fits exist in the top right corner, suggesting we are effectively fitting a standard geometric burst model in these cases. Nonetheless, there are a number of genes for which the parameter sets reside somewhere in the center, indicating that the full complexity of the Γ-OU or CIR models is necessary to describe the corresponding data.

The likelihood ratio procedure yields results that are (i) similar to the distribution shapes observed in the raw data, up to possible numerical errors; (ii) broadly consistent with the predictions from the reduced model fit, although some discrepancies do occur, particularly for 'mixture-like' genes that exhibit higher identifiability under the full model; (iii) qualitatively consistent between datasets; and (iv) largely,

but not perfectly, coherent with a full Bayesian procedure. Therefore, the distributions associated with the proposed models can be distinguished in practice. Further, these differences can be probed using a range of tools, some more approximate and suited to genome-wide exploratory analysis, others more statistically rigorous and suited to detailed study of specific gene targets.

## Discussion

We have introduced a class of interpretable and tractable models of transcription, and characterized the properties of two biologically plausible members of that class. Our results foreground several considerations for experimental design and modeling in modern transcriptomics. Interpretable stochastic models encode mechanistic insights, and motivate the collection of data necessary to distinguish between mechanisms. A variety of stochastic differential equations can describe a variety of biophysical phenomena. Through the methods explored in the current study, they can be coupled to models of downstream processing and used to generate testable hypotheses about RNA distributions. Therefore, our SDE–CME framework can guide experiments to parametrize and distinguish between biologically distinct models of transcription.

The results suggest that single-cell RNA sequencing data may be sufficiently rich to enable Bayesian model discrimination. We analyzed four biological replicates and identified genes that demonstrated visible differences between the two models' fits, which may imply differences in the underlying regulatory motifs. Interpreting the specific biochemical meaning of the findings is challenging without accounting for features which have been omitted in the discussion thus far, such as technical noise and additional complexities in downstream processing of RNA. In Section 9 in the Supplementary Note, we discuss how the models may be extended to account for such phenomena.

The dramatic effect of dynamic contributions suggests that simple noise models need to be questioned. The slow-reversion regime assumed by the mixture model, which presupposes that the evolution of parameters is substantially slower than RNA dynamics, is attractive but potentially implausible. The parameter set we use to illustrate the slow-reversion regime (see Supplementary Table 5) has a noise time scale $\kappa^{-1}$ an order of magnitude longer than degradation, yet still produces distributions that noticeably deviate from the mixture model in their tail regions (Supplementary Figs. 2 and 3). The lifetime of a human mRNA is on the order of tens of hours[74]. Therefore, using a mixture model is formally equivalent to postulating a driving process with an autocorrelation time of weeks. In practice, if the noise time scale is assumed to be on the order of hours to tens of hours, it is useful to explore non-stationary effects, especially if the analysis focuses on tail effects[47]. Our SDE–CME tools facilitate this exploration.

The collection and representation of multimodal data are particularly fruitful directions for experimental design. Even if individual marginals are too similar to use for statistics, joint distributions may be able to distinguish between mechanisms. Aggregating distinct molecular species as a single observable (i.e., modeling the variable $X = X_N + X_M$) neglects biologically important[75–77] regulatory processes of splicing and export buffering. Further, as demonstrated in Fig. 3d, marginal distributions may be insufficiently distinct to identify one of two competing model hypotheses, even with perfect knowledge of the stationary distribution, autocorrelation, and chemical parameters. The bioinformatic barriers to generating full gene-specific splicing graphs based on uncharacterized and infrequent intermediate isoforms are formidable. However, the analytical solutions easily accommodate such data, by solving slightly more complicated versions of the ODEs in Eqs. (14) and (16) (as discussed previously[50] and summarized in Section 9 in the Supplementary Note). Therefore, the deliberate collection of multimodal data is a natural direction for the rational and model-guided planning of high-throughput sequencing experiments.

The identical analytical results for the models' lower moments underscore the need to consider full distributions of molecular species. Although moment-based estimates are useful for qualitative comparisons, and computationally efficient for large bioinformatics datasets, they are insufficient for resolving distinctions even between relatively simple models[68]. Fits to simulated and synthetic data suggest that joint distributions provide considerably more statistical power than marginals.

In studying the Γ-OU and CIR models, we found and validated several distinct asymptotic regimes. Both models recapitulate the constitutive and mixture models in the slow-driving limit ($\kappa$ very small). However, in the limit of bursty production ($\kappa$ large and $\theta$ large), they produce qualitatively different behaviors: the Γ-OU model yields geometric bursts of transcription, whereas the CIR model yields inverse Gaussian driving (see Section 5.3 in the Supplementary Note) with an infinite number of bursts in each finite time interval. We explicitly solved the inverse Gaussian-driven system and computed the generating function, filling an apparent lacuna[78] in the quantitative finance literature. Discrepancies between the models motivate the quantitative investigation of the effects of jump drivers on the molecule distributions, as even this preliminary study shows that they produce drastically different tail behaviors. Further, we identified a fast-mean-reversion, mean-field regime with rapid fluctuations ($\kappa$ very fast), which yields effectively constitutive behavior.

The mathematical methods detailed in the Supplementary Note bear further mention, as they can be substantially generalized. The solution for the Γ-OU model given in Section 3.2 exploits an isomorphism between the CME and the underlying driving SDE[50]. However, this relation is not practical to apply to broader classes of models. As shown in Section 3.3, the path integral method can recapitulate the solution, with robust performance under wider classes of driving processes. More generally, stochastic path integral and physics-inspired methods have recently proven useful for obtaining analytical solutions to relatively complicated stochastic models[45,53,79,80]. As discussed in Section 3.1, we take this opportunity to explore the diversity of solution methods and emphasize useful unifying themes.

Interestingly, certain superficially different models of regulation can be described by the same models. We have motivated the SDEs by endogenous mechanisms, localized to a single cell. However, these models can also describe exogenous variability, such as the transport of regulators into and out of a cell. For example, the mean-reversion term in Eq. (4) can model passive equilibration with an extracellular medium, while the noise term can model active transport into the cell. The form of the noise coarsely encodes the physics of the transport: if a regulator is introduced in bursts (e.g., by vesicle transport), the regulator's concentration can be described by a Γ-OU process, whereas if it is introduced by a constant-rate transporter, its concentration can be described by a CIR process. This interpretation is intriguing in light of extensively characterized gene co-expression patterns observed in cultured cells[81–83]. Inspired by these results, we propose that the toolbox of SDE–CME models can achieve a mechanistic, yet tractable, treatment of co-expression, modeling the concentration of a multi-gene regulator by a continuous stochastic process.

The availability of numerical solvers suggests natural directions for future study. So far, we have treated the case of transcription rates with time-independent parameters at steady state. However, if the parameters vary with time, it is straightforward to adapt the numerical routines to produce full time-dependent distributions for even more general drivers with a combination of stochastic and deterministic effects. This extension provides a route to explicitly modeling the non-stationary behavior of systems with relatively rapid driver time scales, such as differentiation pathways and the cell cycle. Conversely, the stochastic simulations designed for this study can be easily adapted to describe systems with complex phenomena, such as protein synthesis,

reversible binding, and diffusion, which are intractable by analytical approaches in all but the simplest cases.

As we have shown, fine details of transcription—including DNA mechanics and gene regulation—may have signatures in single-cell data, and a model-based, hypothesis-driven paradigm may help identify them. Just as microscopes permit biologists to see beyond their eyes when inspecting a plate of cells, so too can mathematical tools allow them to extract finer insight from the same transcriptomic data.

## Methods

The Supplementary Note contains comprehensive derivations and descriptions of analytical procedures. A complete list of major technical results is presented in Section 2. The Γ-OU and CIR models are fully motivated and solved in Section 3. Moments and autocorrelations are derived in Section 4. Limiting cases are derived in Section 5. Simulation details and validation of our exact results are presented in Section 6. Brief summaries of certain aspects of this work covered more fully in the supplement, and important miscellaneous information, are provided below.

### Notation

A complete guide to our mathematical notation is presented in Section 2.2 in the Supplementary Note. The molecular species of interest are nascent transcripts $\mathcal{N}$ and mature transcripts $\mathcal{M}$. Their respective counts are denoted by random variables $X_N$ and $X_M$. The gene locus produces $\mathcal{N}$ with a time-dependent rate $K(t) = K_t$, described by a stochastic process. Therefore, the probability density of the system is given by $P(X_N = x_N, X_M = x_M, K_t \in [K, K + dK], t)$, i.e., the density associated with finding the system in a state with $x_N$ molecules of $\mathcal{N}$, $x_M$ molecules of $\mathcal{M}$, and a transcription rate of $K$ at time $t$. Having introduced this rather formal notation, we use a shorthand that elides the random variables.

### Model definitions

The Γ-OU and CIR models are mathematically defined via master equations, which describe how probability flows between different possible states. In particular,

$$\frac{dP(x_N, x_M, K, t)}{dt} = \text{CME} + \text{FPE}, \tag{18}$$

$$\text{FPE}_{\Gamma-\text{OU}} = -\frac{\partial}{\partial K}[(a\theta - \kappa K)P] + a\sum_{n=2}^{\infty}(-\theta)^n\frac{\partial^n P}{\partial K^n}, \tag{19}$$

$$\text{FPE}_{\text{CIR}} = -\frac{\partial}{\partial K}[(a\theta - \kappa K)P] + \kappa\theta\frac{\partial^2(KP)}{\partial K^2}. \tag{20}$$

The CME term is identical for both models, and encodes transcription, splicing, and degradation reactions as in the constitutive model[67] (see Section 2 in the Supplementary Note). However, the Fokker-Planck equation (FPE) terms beyond first order, which encode transcription rate variation, are different.

### Analytically solving the Γ-OU and CIR models

The Γ-OU model can be analytically solved using previous results for the $n$-step birth-death process coupled to a bursting gene. This approach exploits the fact that the source species of such a system has a Poisson intensity described by the Γ-OU process, and is fully outlined in Section 3.2 in the Supplementary Note. We set up a system with a bursting gene coupled to a 3-step birth-death process, characterized by the path graph $\varnothing \xrightarrow{a} B \times \mathcal{T}_0 \xrightarrow{\kappa} \mathcal{N} \xrightarrow{\beta} \mathcal{M} \xrightarrow{\gamma} \varnothing$, where $B \sim Geom$ with mean $\theta/\kappa$.

The stochastic process describing the Poisson intensity of $\mathcal{T}_0$ is precisely the Γ-OU process[84]. This implies that the joint distribution of the downstream species coincides with the system driven by Γ-OU transcription. The generating function of SDE-driven system can be

computed using the solution of the bursty system, reported in Eq. (13), where $U_0(s; u_N, u_M) = A_0 e^{-\kappa s} + A_1 e^{-\beta s} + A_2 e^{-\gamma s}$ can be computed by solving Eq. (14):

$$A_2 = u_M\frac{\beta}{\beta - \gamma}\frac{\kappa}{\kappa - \gamma},$$
$$A_1 = \frac{\kappa}{\kappa - \beta}\left(u_N - u_M\frac{\beta}{\beta - \gamma}\right), \tag{21}$$
$$A_0 = -A_1 - A_2.$$

The CIR model is solved using a state space path integral representation of $P(x_N, x_M, K, t)$ which combines a path integral representation of the CME[59] with a more conventional continuous state space path integral. The Γ-OU model can also be solved using this method, along with a plethora of other discrete-continuous hybrid models.

### Analytically computing moments and autocorrelation functions

The master equation satisfied by $P(x_N, x_M, K, t)$ can be recast as a partial differential equation (PDE) satisfied by $\phi(u_N, u_M, s, t)$ (see Section 3 in the Supplementary Note):

$$\frac{\partial\phi}{\partial t} = u_N\frac{\partial\phi}{\partial s} + \beta(u_M - u_N)\frac{\partial\phi}{\partial u_N} - \gamma u_M\frac{\partial\phi}{\partial u_M}$$
$$+ a\theta s - \kappa s\frac{\partial\phi}{\partial s} + f(s), \tag{22}$$

$$f_{\Gamma-\text{OU}}(s) = a\sum_{n=2}^{\infty}\theta^n s^n, \tag{23}$$

$$f_{\text{CIR}}(s) = s^2\kappa\theta\frac{\partial\phi}{\partial s}. \tag{24}$$

By taking certain partial derivatives of the above PDEs, we can recover ODEs satisfied by moments and autocorrelation functions. These can then be straightforwardly solved to compute them.

### Obtaining RNA count distributions from analytical solutions

The aforementioned analytical solutions to each model are in the form of generating functions. To numerically obtain predicted distributions, we first compute the generating function (Eq. (12)) by numerically solving ODEs and integrating the results (i.e., using Eqs. (13) and (14) or Eqs. (15) and (16)). Next, we take an inverse fast Fourier transform[48,85]. To avoid artifacts, the ODEs must be evaluated for a sufficiently fine grid of $g_N$ and $g_M$ on the complex unit sphere.

### Stochastic simulation

Stochastic simulations can verify our analytical results and enable further facile extensions to SDE-driven systems that are otherwise analytically intractable. Because our models involve no feedback, we split this problem into two parts: first, we simulate the continuous stochastic dynamics of the transcription rate $K(t)$, and then we simulate the discrete stochastic dynamics of the nascent and mature RNA using a variant of Gillespie's direct method[86]. This approach requires evaluating reaction waiting times for time-varying transcription rates. For the Γ-OU model, we computed these times exactly *via* the Lambert W function. For the CIR model, we used a trapezoidal approximation to the integral of the reaction flux.

To ensure that all regimes of interest are verified, we chose six parameter sets to test: four of these lie in the extreme limits shown in Fig. 2, and two lie in intermediate regimes. We performed $10^4$ simulations for each parameter set, with $\beta = 1.2$ and $\gamma = 0.7$. The trajectories were equilibrated until a putative steady-state time $T_{ss}$. Afterward, the simulations were left to run until $T_{ss} + T_R$ to enable the computation of autocorrelations. The parameters as well as values of $T_{ss}$ and $T_R$ are

reported in Supplementary Table 5. The implementation details and simulation results are given in Section 6 in the Supplementary Note.

## Data processing
We used four independent mouse datasets generated by the Allen Institute for Brain Science[71,87]. We pseudoaligned the raw reads to a combined intronic/exonic mm10 mouse genome reference using *kallisto|bustools*, yielding spliced and unspliced count matrices[72,88]. We used the default *bustools* filter to remove low-quality cells. To obtain relatively homogeneous cell types, we did not recluster the data. Instead, we used pre-existing cell type annotations, removing all cells with fewer than $10^4$ total molecules.

## Filtering single-cell transcriptomic data
The filtering procedure used data from a single mouse (see Section 8 in the Supplementary Note), and involved the following steps. We selected a series of moderate- to high-abundance glutamatergic cell subtypes (L2/3 IT, L5 IT, L6 IT, L5/6 NP, and L6 CT, taken from sample B08). Genes whose expression levels were too low ($\mu_N, \mu_M \leq 0.01$, or $\max(X_N), \max(X_M) \leq 3$) or too high ($\max(X_N), \max(X_M) \geq 400$) were removed, leaving 3677 genes. Next, to find genes which are potentially well-described by the Γ-OU and CIR models, we fit the three computationally simpler overdispersed limiting models depicted in Fig. 2 using the *Monod* package[73]. Within *Monod*, the *SciPy* implementation of L-BFGS-B was used to perform gradient descent[89] and obtain maximum likelihood estimates for the three-parameter reduced models. We selected genes most consistently assigned to each model (Fig. 4a) according to their Akaike weights[90]. This step identified genes that appeared to be reproducibly described by each model class, and provided a tentative basis for out-of-sample predictions. Finally, we restricted our analysis to the best-fit 35 genes in each category, as quantified by the maximum rank of the chi-squared statistic observed across the five subtype datasets. This filtering step was applied to avoid contributions due to model misspecification or poor convergence, and focus on the genes that best agreed with the regimes of interest. The preliminary analysis produced 35 genes of interest for the Γ-OU-like and CIR-like categories and 10 genes for the mixture-like category.

## Fitting SDE−CME models to simulated and single-cell transcriptomic data
The *SciPy* implementation of L-BFGS-B was used to perform gradient descent[89] and obtain maximum likelihood estimates for the four-parameter SDE−CME models. To control for potential failure to converge, we omitted all results with log-likelihood ratios with magnitude above 150 from visualization in Fig. 4. All fits to raw data are shown in Supplementary Figs. 8–37. Fits with large likelihood ratios typically corresponded to poor fits to one or both of the models, possibly due to numerical issues. The Python package *PyMC3* was used to sample the parameter posteriors, using the non-gradient-based Markov chain Monte Carlo sampler DEMetropolisZ[91]. Synthetic data inference used a uniform prior, four chains, 1000 burn-in iterations, and 12,000 sampling iterations. Biological data inference used a uniform prior, one chain, and 1000 iterations.

## Reporting summary
Further information on research design is available in the Nature Portfolio Reporting Summary linked to this article.

## Data availability
Publicly available data were downloaded from the NeMO archive. The metadata were obtained from http://data.nemoarchive.org/biccn/grant/u19_zeng/zeng/transcriptome/scell/10x_v3/mouse/processed/analysis/10X_cells_v3_AIBS. Raw FASTQs were obtained from http://data.nemoarchive.org/biccn/grant/u19_zeng/zeng/transcriptome/scell/10x_ v3/mouse/raw/MOp/. Pre-built genome references were obtained from the 10× Genomics website, at https://support.10xgenomics.com/single-cell-gene-expression/software/downloads/latest. The FASTQ files were used to generate loom files with spliced and unspliced count matrices. These count matrices are available in the Zenodo package 10.5281/zenodo.7262328[92]. The results of the fits generated with the Monod package, the SDE gradient descent fit, and the MCMC fit are available at https://github.com/pachterlab/GVFP_2021, as well as the Zenodo package 10.5281/zenodo.7262328[92]. All synthetic data, generated using custom stochastic simulation code, as well as the simulation parameters, are deposited in the GitHub and Zenodo repositories[92].

## Code availability
Single-cell RNA sequencing data were pseudoaligned using *kallisto|bustools* 0.26.0, wrapping *kallisto* 0.46.2 and *bustools* 0.40.0. Dataset filtering, reduced model fits, and Akaike information criterion computation were performed using *Monod* 0.2.4.0. MCMC parameter inference was performed using *PyMC3* 3.11.4, dependent on *Theano-PyMC* 1.1.2. Data input/output were performed using *loompy* 3.0.7. Numerical procedures, such as gradient descent and quadrature, were performed using *SciPy* 1.4.1 and *NumPy* 1.21.5. The algorithms were implemented in the framework of Python 3.7.12. All code is available at https://github.com/pachterlab/GVFP_2021 and the associated Zenodo package 10.5281/zenodo.7262328[92]. The GitHub and Zenodo repositories include scripts used to construct a mouse genome reference, pseudoalign datasets, and generate all figures. They are modular: the analysis can be restarted at a set of intermediate steps. The outputs of certain steps, viz. pseudoaligned count matrices, results of the Monod pipeline, the list of genes of interest, results of the gradient descent procedure, and results of the Bayes factor computation procedure can be recomputed, or loaded in based on files available in the repositories. Synthetic data generated by simulation, as well as the routines used to generate the data, are available in the repositories. The CIR simulation is implemented in Python 3.7.12. The Gamma-OU simulation was developed using MATLAB 2020a, and executed in the Python wrapper for Octave, using versions *oct2py* 5.4.3 and *octave-kernel* 0.34.1.

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

## Acknowledgements

The DNA, pre-mRNA, and mature mRNA used in Fig. 1 are derivatives of the DNA Twemoji by Twitter, Inc., used under CC-BY 4.0. G.G. acknowledges the help of Victor Rohde in exploration of the stochastic process literature. G.G., M.F., and L.P. were partially funded by NIH U19MH114830. J.J.V. was supported by NSF Grant # DMS 1562078.

## Author contributions

J.J.V. and G.G. conceived of the work, derived the mathematical results, and drafted the manuscript. G.G., M.F., and J.J.V. worked on simulating the models and numerically implementing their analytic solutions. G.G. and M.F. fit the single-cell data. L.P. supervised the work. All authors reviewed and edited the manuscript.

## Competing interests

The authors declare no competing interests.
