## [Peer Review File · Nature Communications]

Interpretable and tractable models of transcriptional noise for the rational design of single-molecule quantification experimentsEditorial Note: Parts of this Peer Review File have been redacted as indicated to maintain the confidentiality of unpublished data.

REVIEWER COMMENTS

Reviewer #1

The manuscript under review presents a class of stochastic models of transcription driven by stochastic time-varying transcription rates. The aim of the authors is to unify and generalise known transcription models characterising cell-to-cell variability of transcription rates as well as to propose ways to use mechanistic models of transcription to motivate biological hypotheses. More specifically, two example models from the proposed class are described where the incorporated transcription rate variation arises from specific assumptions about the DNA mechanics and the regulation of gene expression. These mechanistic assumptions lead to two candidate mathematical models which couple the discrete reaction kinetics of synthesis, splicing and degradation of mRNA, governed by the CME, with the continuous stochastic dynamics of the transcription rate, governed SDEs. The authors conduct a thorough mathematical analysis to explore the effects of these two models directly on mRNA synthesis dynamics as well as on the predicted transcriptomic count distributions.

The authors provide full analytical solutions of the proposed stochastic models. From these solutions, well studied limiting distributions are recovered, as well as a novel class of distribution arises, which is characterised as quasi-bursty behaviour. A central result is that the two studied models produce identical steady-state summary statistics (upto the second moments). The authors use this result to conclude that summary statistics are not sufficient to distinguish between competing underlying models of transcriptomic count data. Further, the authors use synthetic data to perform Bayesian model comparison. Precisely, they use Bayes factors to show how distinguishable are the two models, for various parameter regimes, as well as how much do the two models differ from constitutive and mixture steady-state models. The authors emphasise the importance of using this class of dynamical models to distinguish between different biological mechanisms of transcription and motivate multimodal experiments that can parametrise this type of models.

Overall, the reported study is a significant body of work that provides new directions towards the development of effective models for studying stochastic gene expression and cell-to-cell variability in transcription mechanisms. The authors successfully convey the main objective and idea, that interpretable and tractable models can guide the generation of biological hypotheses and the design of experiments that can test those hypotheses. In the comments below, some weaknesses of the paper are highlighted, and recommendations are provided for improving the structure and the presentation of results. Provided that the authors pay attention to these details, the manuscript should definitely be considered for publication.

Comment on originality:

- The proposed stochastic models of transcription have not been reported previously in relevant studies. The specific class of models that couple the CME description of gene expression with the Γ -OU and CIR SDEs driving the synthesis rates are novel to the best of my knowledge. However, a relevant piece of work that in my opinion is neglected and should be cited by the authors is the article "Stochastic models of gene transcription with upstream drives: exact solution and sample path characterization", J. Dattani and M. Barahona, J. R. Soc. Interface 14:20160833.

- Some of the main claims about the advantages of using the full distributions to moments only to infer gene expression is not that novel. It has been long known issues regarding non-identifiability of gene expression parameters and mechanism using gene expression noise data. For example, see early work by Michael Stumpf and Johan Paulsson groups. A main challenge of using subtle differences in

the distributions to uncover mechanisms is model misspecification. The real data will have specific technical errors. The current models ignore many things such as the effect of cell growth and division on distributions of molecules (see for example recent work by Ramon Grima and Philipp Thomas on this topic, e.g. Thomas and Shahrezaei Journal J. R. Soc. Interface 2021).

- Although in the last section of the Results, the authors talk about "Bayesian inference" and "parameter inference", they haven't fitted the models to real single-cell RNA-seq data. All the validations/comparisons of the models and the inference are based in synthetic data scenarios. In my opinion, a great addition to this study would be an attempt to infer parameters of the predicted whole count distributions using real single-cell nascent and mature count data for example from scRNA-seq data. This will highlight the practical limits of such theoretical proposals.

Comments about the study and the main text:

- The authors motivate well the use of a stochastic time-varying transcription rate to account for observed variability in transcription, as a more effective modelling assumption than just assuming a steady-state distribution of transcription rates to account for cell-to-cell variability. However, they do not clearly justify why their models account for variability only in the transcription rate and not in the splicing & degradation kinetics of gene expression.
- The authors motivate well the two assumed production rate mechanisms and the SDEs driving the stochastic transcription. One point that could be improved in the paragraph where the authors introduce the Γ - OU model, is the following sentence: "The scaling between the relaxation state and the transcription rate is set by a gain parameter θ ". This sentence does not make very clear what is the biological meaning of the parameter θ . Perhaps it would be useful if the authors included an illustration / cartoon which visualises the reaction mechanism that they introduce (especially if they want the characterisation "gain" to make sense). Another unclear point in the paragraph about the CIR model is the sentence "Because the SIR model remains agnostic about...used to represent more biophysically detailed hypotheses". In my opinion this argument lacks ground and does not justify the effectiveness of the proposed CIR model. Also, similarly a figure illustrating how the CIR production mechanism works would aid the reader to understand better.
- In the paragraph with title "The models are interpretable and unify known results", it would be good if the authors emphasised more on the effects of intrinsic and extrinsic contributions to noise. Although they show the derived formulas for the CVs of nascent and mature mRNA counts, it is not made clear how intrinsic and extrinsic noise affects the transcript levels. Perhaps it would be useful if the authors include a group of figures illustrating how the levels of noise change for certain interesting parameter regimes (for example as they mention, in the "overdispersed regimes").
- Some of the paragraphs in the section "Models can be distinguished using multimodal count data" have space for improvement. In the sentence "This suggests that, while probabilistic inference...many naïve approaches may fail", what exactly is meant by naïve approaches? A bit further, after the sentence "To simplify the computation, we assumed that β , γ and $\langle K \rangle$ are known, because they can be accurately and robustly inferred from empirical means", the authors could justify further why the inference of this parameters is accurate and robust. More generally, it feels that this section of the paper is written in a rushed and "sloppy" way. Although the results are rich and they can convey a lot of information, the authors have not managed to give a detailed and crystal-clear presentation of these results. In my opinion, especially a biologist or a scientist with not a strong mathematical background might struggle to appreciate this section of results. Especially because the main goal of this manuscript is to highlight the role of mathematical models in motivating the design of experiments, it is important that the authors clearly communicate how models can be distinguished from data as well as how experiments can be used to parametrise these models.

- The Discussion section is written more carefully and clearly and in my opinion the authors should use a similar approach on how they present their results in order to restructure and strengthen some paragraphs in the Results subsections.
- The authors propose a method for doing Gillespie simulations with time varying parameters. There are several algorithms in the literature for such tasks (for example see Shahrezaei et al Mol Sys Biol 2008 and Voliotis et al PLoS Comp Biol 2016). How does their proposed approach related to the previous methods.

Reviewer #2

Summary:

In their manuscript, Gorin et al. discuss mathematical models based on stochastic differential equations and chemical master equations (SDE-CME) addressing transcription rate variation. The authors argue that the analysis of models that are tractable and interpretable can help in identifying the most discriminative data between competing model hypotheses and, hence, in gaining mechanistic understanding of the molecular processes underlying available transcriptomic data.

In accordance with empirical observation, RNA count distributions are often assumed to be negative binomially distributed. The authors point out that this assumption lacks direct interpretability and mechanistic insight with respect to the source of single-cell transcription rate variation. Hence, they analyze two alternative mathematical models, both tractable and interpretable: the Gamma Ornstein-Uhlenbeck (Γ -OU) model, assuming DNA stress and stress-relieving topoisomerases to underlie transcription rate variation, and the Cox-Ingersoll-Ross (CIR) model, assuming local density fluctuations of regulatory molecules responsible for transcription rate variation. Mathematical analysis reveals that both models are governed by different probability distributions but exhibit the same summary statistics.

Using a simulation study, the authors show that (1) multimodal data (meaning nascent + mature RNA counts) enables better discrimination between the two models, (2) the Γ -OU and CIR models are well distinguishable from simpler models (Poisson and static mixture models), and (3) the nascent and mature RNA count distributions of both models for maximally divergent parameter sets are still visually similar. Finally, Gorin et al. show that the model parameters κ and θ of the Γ -OU model can only be reliably inferred for strongly non-Poisson cases.

The manuscript is nicely structured, well written, provides detailed methods, and is easy to follow.

We acknowledge that the mechanistic models for gene expression discussed and solved in the manuscript are definitely noteworthy, and that model inference on simulated data is interesting. However, to have a considerable impact on the field, we think that the proof-of-concept application to experimental data is mandatory to show the practical scope of the method. This is unfortunately missing in the current manuscript.

* Major issues:

Significance:

- For broad impact, the authors should showcase how their approach can be applied to experimental data to discriminate between the two proposed models (Γ -OU and CIR).
- If the approach cannot be applied to existing experimental data, the authors should elaborate on how a suitable experiment should be designed.
- While the authors clearly state in lines 42-43: "While we will not actually implement the entire closed loop paradigm depicted in Figure 1a, our work constructs one possible mathematical and computational foundation for it.", we would have loved to see the example presented by Gorin et al. to be taken further, e.g., by showing that experimentally measured nascent and mature RNA count distributions can be better described by either the Γ -OU or CIR model in comparison to the standard

negative binomial distribution assumption.

Novelty:

- In the abstract the authors write “We argue that answering this question requires quantitative models that are both interpretable (describing concrete biophysical phenomena) and tractable (amenable to mathematical analysis”. The idea of using mathematical analysis and simulations to identify whether one can discriminate between competing model hypotheses however is not novel, and a pillar of systems biology.
- In a recent preprint (Direct simulation of a stochastically driven multi-step birth-death process; <https://doi.org/10.1101/2021.01.20.427480>) Gorin and Pachter present a detailed discussion of the Γ -OU model and a simulation method for the corresponding SDE-CME. This limits the novelty of the present manuscript. Also, an explicit reference to this work of the same authors is missing.
- In the abstract the authors write: “We introduce a framework to numerically generate these predictions and analytically unify known transcriptional regimes.” Is the simulation method that the authors refer to the same as the one that has been put forward in the preprint?
- The authors should more strongly elucidate the novelty of current study with respect to Ref. 80 and emphasize its significance to the field.

* Minor issues:

Wording/Text:

- Please define the term ‘mean reversion’ in equation (4) similar to its description in caption Figure 1b.
- In line 125: The acronym CLE is not defined.

Presentation:

- The upper points in Figure 1a “Hypothesis-driven transcriptomics experiment” and “model rejection be incorporated” are not addressed in the manuscript, which is touched on in the main text but might not become apparent from cross-reading the paper.
- Figure 1c: We suggest to highlight that the dots in the schematic describing the Γ -OU model represent the stress-relieving topoisomerases, while the same dots in the schematic describing the CIR model represent regulatory molecules.
- Figure 1c: What do the colors represent? E.g. orange stress arrow?, pink dot?, green relief arrow?, orange DNA coil? If important, please add a legend or labels to the figure. Same holds for the arrows in Figure 1b.
- Eq. 1: Please mention why decay of nascent RNA is not considered here.
- Line 83: Has ‘constitutive model’ been defined?
- Eq. 6: Please make $\epsilon(t)$ more explicit by emphasizing that a and θ enter here.
- Figure 2b: Are the plots shown schematics or results of simulations? Please specify.
- Figure 2b: Please state somewhere in the figure caption that the green shades of schematics $P(K,t_1)$ and $P(K,t_2)$ represent the values of K .
- Figure 2b: The schematics of $P(K,t_1)$ and $P(K,t_2)$ for Γ -OU and CIR look bimodal, which does not fit the respective schematics for $K(t)$. Please adapt the green colors in the schematics $P(K,t_1)$ and $P(K,t_2)$ to show a more representative image, maybe similar to your tweetorial 7/ <https://twitter.com/lpachter/status/1436344431657979913>.
- Figure 2b: The schematics of $P_{ss}(x_N)$ of the mixture model and Γ -OU look identical. However, they should have different probability distributions for nascent RNA counts. Please adapt if only schematics and no simulations.
- In lines 143 and 144 you state that: ‘However, the CIR model predicts a novel family of count distributions with heavier tails than their Γ -OU counterparts.’ This is not easily visible in Figure 2b. We suggest to graphically highlight the differences between the $P_{ss}(x_N)$ schematics for Γ -OU and CIR more strongly.
- You only show the probability distribution of nascent RNA counts in Figure 2b. As you also identify differences between the processes in the mature RNA compartment, one could already allude to this in

Figure 2b as an extra column $P_{ss}(x_M)$.

- Figure 3a: How different do the results look for a different set of ground truth parameters $\langle K \rangle$, β , and γ as defined in line 189? Given that the interpretability of the models plays a crucial role, is there a specific reason to focus on the ground truth parameters defined in line 189?
- Figure 3a: Please show the color legend as a gradient scale with corresponding $\log_{10}(\text{BF})$ values.
- Please add the axis title $\log_{10}(\text{BF})$ in Figure 3c for clarity.
- In lines 205 and 206 the authors state that ‘... and found that even when predictions are maximally divergent, the distributions are still visually alike (Figure 3d).’ Is a statistical test able to reject the hypothesis that the probability distributions of the nascent and mature RNA counts for Γ -OU and CIR are from the same underlying distribution?
- Figure 3e: Do these results also hold for the CIR model?
- Figure 3e: The middle plots showing the nascent versus the mature RNA counts for an overdispersed and Poisson-like case are distracting. If the authors find it important to visually exemplify this, these plots could be moved to Figure 2b, where the models and their probability distributions are introduced.
- Figure 3e: Please add a colorbar.
- In lines 211 and 212 it is stated that “To simplify this computation, we assumed that β , γ , and $\langle K \rangle$ were known, because they can be accurately and robustly inferred from empirical means (one can also imagine performing separate experiments to determine them first)”. How would the results from Figure 3e look like if the parameters β , γ , and $\langle K \rangle$ were indeed obtained from simulated data?

Reviewer responses

Reviewer #1 (Remarks to the Author):

The manuscript under review presents a class of stochastic models of transcription driven by stochastic time-varying transcription rates. The aim of the authors is to unify and generalise known transcription models characterising cell-to-cell variability of transcription rates as well as to propose ways to use mechanistic models of transcription to motivate biological hypotheses. More specifically, two example models from the proposed class are described where the incorporated transcription rate variation arises from specific assumptions about the DNA mechanics and the regulation of gene expression. These mechanistic assumptions lead to two candidate mathematical models which couple the discrete reaction kinetics of synthesis, splicing and degradation of mRNA, governed by the CME, with the continuous stochastic dynamics of the transcription rate, governed SDEs. The authors conduct a thorough mathematical analysis to explore the effects of these two models directly on mRNA synthesis dynamics as well as on the predicted transcriptomic count distributions.

The authors provide full analytical solutions of the proposed stochastic models. From these solutions, well studied limiting distributions are recovered, as well as a novel class of distribution arises, which is characterised as quasi-bursty behaviour. A central result is that the two studied models produce identical steady-state summary statistics (upto the second moments). The authors use this result to conclude that summary statistics are not sufficient to distinguish between competing underlying models of transcriptomic count data. Further, the authors use synthetic data to perform Bayesian model comparison. Precisely, they use Bayes factors to show how distinguishable are the two models, for various parameter regimes, as well as how much do the two models differ from constitutive and mixture steady-state models. The authors emphasise the importance of using this class of dynamical models to distinguish between different biological mechanisms of transcription and motivate multimodal experiments that can parametrise this type of models.

Overall, the reported study is a significant body of work that provides new directions towards the development of effective models for studying stochastic gene expression and cell-to-cell variability in transcription mechanisms. The authors successfully convey the main objective and idea, that interpretable and tractable models can guide the generation of biological hypotheses and the design of experiments that can test those hypotheses. In the comments below, some weaknesses of the paper are highlighted, and recommendations are provided for improving the structure and the presentation of results. Provided that the authors pay attention to these details, the manuscript should definitely be considered for publication.

Comment on originality:

- The proposed stochastic models of transcription have not been reported previously in relevant studies. The specific class of models that couple the CME description of gene expression with the Γ -OU and CIR

SDEs driving the synthesis rates are novel to the best of my knowledge. However, a relevant piece of work that in my opinion is neglected and should be cited by the authors is the article “Stochastic models of gene transcription with upstream drives: exact solution and sample path characterization”, J. Dattani and M. Barahona, J. R. Soc. Interface 14:20160833.

We thank the reviewer for directing our attention to Dattani and Barahona’s article, which is indeed closely related. Just as in our work, one considers the count distributions induced by a transcription process driven by a (possibly random) upstream driver which adds extra variability, possibly in a way that correlates the transcription of different cells in one population. They also derive nice analytical results that we are eager to further explore, and that could be used to extend our work.

One sense in which we build on the work of researchers like Dattani and Barahona is that we consider $K(t)$ and its distribution not to be known a priori, but determined dynamically according to an SDE. In this way, we treat $K(t)$ as a continuous stochastic random variable on the same footing as nascent and mature RNA counts. To emphasize this, we have added the following text to the end of “Interpretable and tractable modeling framework...”:

“Coupling upstream variability to transcriptional CMEs has been studied before (e.g. by Dattani and Barahona, but usually in a way that assumes either that $K(t)$ takes on a finite set of values (for example, gene switching), or that the distribution of $K(t)$ is a priori known, rather than defined by a stochastic dynamical system like Eq. 4. We attempt to build on these studies by treating $K(t)$ as a continuous stochastic dynamical variable on the same footing as nascent and mature RNA counts.”

Although researchers like Dattani and Barahona have considered coupling transcription to upstream drivers, we consider this difference to be important in a technical as well as conceptual sense, given that the SDE assumption leads to nontrivial mathematical structure that makes solving and simulating the models we consider to be a difficult task.

- Some of the main claims about the advantages of using the full distributions to moments only to infer gene expression is not that novel. It has been long known issues regarding non-identifiability of gene expression parameters and mechanism using gene expression noise data. For example, see early work by Michael Stumpf and Johan Paulsson groups.

A main challenge of using subtle differences in the distributions to uncover mechanisms is model misspecification. The real data will have specific technical errors. The current models ignore many things such as the effect of cell growth and division on distributions of molecules (see for example recent work by Ramon Grima and Philipp Thomas on this topic, e.g. Thomas and Shahrezaei Journal J. R. Soc. Interface 2021).

We agree that model misspecification poses a considerable risk, and added the following text in the Results section to emphasize this:

“Even if the Gamma-OU and CIR models can be distinguished and fit to data in principle, can they be distinguished and fit in practice? Real transcriptomic data feature additional noise due to technical errors, and possibly confounding influences due to phenomena like cell growth and division. One can also face serious model misspecification problems, where one finds that even though one model fits better than others, none of them fit particularly well.”

We also cited the Thomas and Shahrezaei article mentioned above. With some caution in mind, we proceed nonetheless, and propose two approaches for minimizing these sources of variation and uncertainty.

The first approach consists of experimental control or judicious data selection. For example, if we are interested in reducing contributions from cell growth and division, we may sequence a dataset from a tissue with terminally differentiated cells, such as blood or the brain. This is the strategy we use in the analysis of scRNA-seq data: we focus on a single type of adult mouse glutamatergic cells, treating them as stationary or ergodic. As such neural cell populations do not typically divide, this approximation appears to be consistent with known biology. Our attempts to control for misspecification are described in the beginning of the section titled “Multimodal count distributions...”

The second approach consists of extending the current models to account for such phenomena. Although we consider implementing them beyond the scope of this paper, we discuss such extensions in Section S9. We find that certain simple models of variability, such as imperfect observation of molecules, non-equilibrium states, and cell type heterogeneity, can be self-consistently incorporated into our framework by modifying the generating function. At this time, the cell cycle resists analogous analysis. It is straightforward to model binomial molecule partitioning, and we refer readers to previous work for a generating function-based exposition of this phenomenon (Gorin and Pachter, *Biophysical Journal* 121:6, 1056-1069, 2022). However, the behavior of $K(t)$ at the moment of division is unclear, and we speculate that the models’ assumptions are violated during the considerable nuclear remodeling that accompanies cell replication.

The approaches described above can work in concord. For example, it may be possible to identify the parameters of technical noise models using a stationary cell population (in the spirit of Gorin and Pachter, *bioRxiv*, 10.1101/2021.07.30.454514v1, 2021), treat these parameters as constant between experiments, and use them as plug-in estimates to fit more sophisticated models that include transient phenomena instead of attempting to learn all parameters simultaneously.

- Although in the last section of the Results, the authors talk about “Bayesian inference” and “parameter inference”, they haven’t fitted the models to real single-cell RNA-seq data. All the validations/comparisons of the models and the inference are based in synthetic data scenarios. In my

opinion, a great addition to this study would be an attempt to infer parameters of the predicted whole count distributions using real single-cell nascent and mature count data for example from scRNA-seq data. This will highlight the practical limits of such theoretical proposals.

We have substantially revised the Results section of the manuscript, applying the procedure described in Figure 3 to identify the models that most effectively recapitulate scRNA-seq data. We perform this comparison for four biological replicates, with data from a single mouse neuron cell type that appears to concord with our biological assumptions of homogeneity and stationarity.

In our analysis, we find that it is possible to identify genes as better described by one or the other model with a high level of statistical confidence. To facilitate exploratory analysis on a genome-wide scale, we implemented an *ad hoc* filtering procedure. From Figure 3a, we know that the models are best distinguishable in the high-gain regime. Therefore, to identify genes which are likely to be Γ -OU-like, CIR-like, or mixture-like (not easily distinguishable), we fit the three reduced models that emerge in the high gain and slow mean-reversion limits -- i.e., the bursty model, the quasi-bursty model, and the mixture model. Maximum likelihood estimates (MLEs) of their parameters can be obtained by gradient descent in three dimensions. By evaluating Akaike information criteria at the MLEs, we select genes which were consistently assigned to a particular limit for five subsampled cell populations in a single mouse sample. The inference procedures are described in Section S8, and are implemented as native functions of the *Monod* package.

Next, we fit the full models using gradient descent in four dimensions on out-of-sample datasets and found that the qualitative predictions were largely recapitulated, with putative Γ -OU-like genes typically having likelihood ratios in favor of the Γ -OU model, putative CIR-like genes having likelihood ratios in favor of the CIR model, and putative mixture-like genes having likelihood ratios near unity and not consistently favoring either of the models. Therefore, although the *ad hoc* filter based on limiting regimes is not perfect, it is qualitatively consistent and computationally facile, providing a useful starting point for analysis. Further, we found that the models favored by the likelihood ratios tended to be consistent between four biological replicates, suggesting that the differences in model assignments correspond to biological mechanisms shared between organisms.

In a subset of the fits, the gradient descent procedure failed to achieve an acceptable fit to data at the maximum likelihood estimate (MLE). We discarded these cases. To ensure the likelihood ratios identified by the gradient descent-based procedure reflect real differences rather than incidental failure to converge, and do not grossly overestimate confidence in a particular model, we fit a subset of twelve genes using Monte Carlo Markov Chain sampling of the parameter posterior and computed Bayes factors from these fits' marginal likelihoods. The results were quantitatively consistent with the MLE-based procedure, suggesting that, with the current workflow and gene selection, the thresholds are sufficiently strict to compensate for potential

errors in optimization, and the contribution of uncertainty beyond the MLE is minimal. The results of the procedure are displayed in Figure 4, which illustrates the selection procedure in panel a, visualizes the out-of-sample likelihood ratios and Bayes factors in panel b, demonstrates a set of illustrative successful fits in panel c, and displays the parameter estimates in panel d.

In sum, our analysis provides a route to identify model differences in real data by implementing and evaluating the following key points:

1. Genes that better match one or the other model can be efficiently identified by comparing fits of the corresponding reduced models, suggesting that such investigations can scale to transcriptome-wide data.
2. When compared *via* likelihood ratios computed from the full models, biological replicates yield consistent results, suggesting promising targets for investigation in future experimental studies.
3. The likelihood ratios are quantitatively consistent with the Bayes factors, suggesting that, as long as sources of numerical error are carefully controlled, the point estimate-based procedures produce results representative of more computationally intensive and statistically rigorous Bayesian analyses.

Comments about the study and the main text:

- The authors motivate well the use of a stochastic time-varying transcription rate to account for observed variability in transcription, as a more effective modelling assumption than just assuming a steady-state distribution of transcription rates to account for cell-to-cell variability. However, they do not clearly justify why their models account for variability only in the transcription rate and not in the splicing & degradation kinetics of gene expression.

We have omitted the discussion of dynamic variability in the splicing and degradation due to three considerations. Firstly, such models are typically intractable; if the rates randomly vary in time, the partial differential equation characteristics used to derive the solutions are no longer well-defined. Formally, treating such a system requires solving a CME coupled to several FPEs, which is considerably more challenging than the current transcriptional variation model.

Secondly, the modulation of downstream reaction rates is not typically invoked in models of transcriptional variation, apparently due to a paucity of evidence for this mechanism. For example, a study of the coupling between cell volume and extrinsic noise magnitude (Padovan-Merhar et al., *Molecular Cell* 58, 339-352, 2015) did not find that degradation played a major regulatory role in this phenomenon. In a recent investigation, we fit a conventional bursty model (Gorin and Pachter, *bioRxiv*, 10.1101/2022.06.11.495771v1, 2022) to identify patterns of biophysical parameter modulation between neuron cell types, and found that apparent differences in (non-dimensionalized) splicing or degradation rates could overwhelmingly be attributed to burst frequency modulation (Figure 4 of the preprint).

Finally, the few studies treating the effects of variation in decay rates (e.g., Baudrimont et al., *Cell Reports* 26, 3752-3761, 2019) have typically adopted a moment-based approach. Therefore, treating variation in spliceosome or RNase concentrations would require developing new, dedicated stochastic models, whose appropriate functional form is not yet clear; for example, the assembly of the spliceosome requires several complex formation steps, so a CIR-like model may not be appropriate to describe its formation and turnover.

As discussed in Section S9, we can, in principle, begin to represent variability in splicing degradation kinetics by defining a generating function mixture (in the spirit of Gorin and Pachter, *Biophysical Journal* 121:6, 1056-1069, 2022, Eq. 40). However, such mixture representations are mostly formal and cannot be easily computed, and only hold in the limit of very slow ribonuclease or spliceosome turnover. Further, our preliminary fits to scRNA-seq data do not seem to point to major sources of unexplained variation. Therefore, we omit discussion of this phenomenon, although we hope that the outlined mathematical tools can be used to treat it in the future.

- The authors motivate well the two assumed production rate mechanisms and the SDEs driving the stochastic transcription. One point that could be improved in the paragraph where the authors introduce the Γ – OU model, is the following sentence: “The scaling between the relaxation state and the transcription rate is set by a gain parameter θ ”. This sentence does not make very clear what is the biological meaning of the parameter θ . Perhaps it would be useful if the authors included an illustration / cartoon which visualises the reaction mechanism that they introduce (especially if they want the characterisation “gain” to make sense).

We adopt the term “gain” as a useful shorthand from the field of chemical engineering process control (e.g., Seborg et al., *Process Dynamics and Control*, Wiley, 2017), where the term is a constant scaling factor between an input and an output of a unit operation or a controller. For example, a reaction engineer may model the temperature change in the reactor as a linear function of the heat input; in that case, the gain is the heat capacity.

In our case, we define the gain as a scaling factor that can describe a variety of possible rapid processes. One such interpretation is RNA polymerase (RNAP) binding: if we suppose that RNAP is present at concentration $\langle p \rangle$ and its turnover dynamics can be neglected, we may define a gain equal to $A\langle p \rangle$. The term A defines the relationship between relaxation $y(t)$ and the polymerase binding rate $k_{ini}(t)$, such that $k_{ini}(t) = A y(t)$. This linear proportional relationship is, in turn, a first-order Maclaurin expansion of the dynamics around $k_{ini} = y = 0$. We propose that this proportional relationship is appropriate for typical eukaryotic systems that demonstrate high levels of chromatin compaction; we expect their dynamics to be restricted to a low-activity region.

We agree that a better illustration would be helpful. To clarify the biophysical intuition underpinning the models, we have modified Figure 1. We have also revised the body of the

manuscript to explicitly discuss the role of RNAP, and extended the discussion in Sections S3.2.1 and S3.3.1.

Another unclear point in the paragraph about the CIR model is the sentence “Because the SIR model remains agnostic about...used to represent more biophysically detailed hypotheses”. In my opinion this argument lacks ground and does not justify the effectiveness of the proposed CIR model. Also, similarly a figure illustrating how the CIR production mechanism works would aid the reader to understand better.

We again agree that a better illustration would be helpful, and modified Figure 1 to clarify the biophysical intuition underpinning the CIR model. To simplify this discussion, we have removed the discussion of the “generic” model $R \rightarrow R+N$. Now, the body of the text focuses on the concrete case of a regulator with time variation in its concentration. We discuss more general cases with the same mathematical form, and their implications for the meaning of the gain factor, in Section S3.3.1.

- In the paragraph with title “The models are interpretable and unify known results”, it would be good if the authors emphasised more on the effects of intrinsic and extrinsic contributions to noise. Although they show the derived formulas for the CVs of nascent and mature mRNA counts, it is not made clear how intrinsic and extrinsic noise affects the transcript levels. Perhaps it would be useful if the authors include a group of figures illustrating how the levels of noise change for certain interesting parameter regimes (for example as they mention, in the “overdispersed regimes”).

One unexpected feature of these models is that the ‘shape’ of a distribution—which depends only on where the corresponding parameter set resides in the two-dimensional qualitative regimes space depicted in Figure 2—is not very closely related to how much of the overall noise is intrinsic or extrinsic.

In particular, define the ‘fraction’ of extrinsic noise to be the extrinsic CV term divided by the total CV (for either nascent or mature RNA). Our analytic results can be used to show that (for example) the fraction of extrinsic noise for nascent RNA is

$$\frac{\theta}{\theta + \kappa + \beta}.$$

This relates to distribution shape somewhat, since for the Poisson-like limiting regimes ($\theta \rightarrow 0$, $\kappa \rightarrow 0$, with the ratio held constant; or $\theta \rightarrow \infty$, $\kappa \rightarrow \infty$, with the ratio held constant) this fraction goes to zero.

But for the overdispersed limits, which limit is ‘noisier’ turns out to be highly parameter-dependent. In the high gain limit ($\theta, \kappa \rightarrow \infty$ with ratio held constant) one obtains $\theta/(\theta + \kappa)$, whereas in the slow reversion limit ($\theta, \kappa \rightarrow 0$ with ratio held constant) one obtains $\theta/(\theta + \beta)$. Which one is bigger depends on the value of β , which is not fixed by the qualitative regimes space or $\langle K \rangle$.

To convey this unexpected point, the following text was added to “The models are interpretable...”:

“In the ‘overdispersed’ regimes, where θ is large or κ is small, the extrinsic noise contributions become significant, but not in a way that maps cleanly onto the space depicted in Figure 2. For example, the fraction of extrinsic noise for nascent RNA is ... whose relative size in different overdispersed regimes changes depending on the splicing rate β .”

Additional technical detail was provided in the SI, in Section S2.4.2. We agree that an illustration of this could be useful, and added Figure S5, which plots the functional dependence of the extrinsic noise fraction.

- Some of the paragraphs in the section “Models can be distinguished using multimodal count data” have space for improvement. In the sentence “This suggests that, while probabilistic inference...many naïve approaches may fail”, what exactly is meant by naïve approaches?

We have appended the following clarification to that section:

“This suggests that, while probabilistic inference using whole distributions may succeed at performing model discrimination, many naïve approaches, such as those based on moments or single marginals, may fail.”

A bit further, after the sentence “To simplify the computation, we assumed that β , γ and $\langle K \rangle$ are known, because they can be accurately and robustly inferred from empirical means”, the authors could justify further why the inference of these parameters is accurate and robust. More generally, it feels that this section of the paper is written in a rushed and “sloppy” way. Although the results are rich and they can convey a lot of information, the authors have not managed to give a detailed and crystal-clear presentation of these results. In my opinion, especially a biologist or a scientist with not a strong mathematical background might struggle to appreciate this section of results.

Especially because the main goal of this manuscript is to highlight the role of mathematical models in motivating the design of experiments, it is important that the authors clearly communicate how models can be distinguished from data as well as how experiments can be used to parametrise these models.

We agree that the original manuscript could have been more clear here, and have completely rewritten this section both to (i) improve clarity, and (ii) remove unnecessary technical detail so that the text is somewhat friendlier to more experimentally-oriented readers. One thing we did to improve clarity was to offer additional motivation for our choices. For example:

“Even if our two models did not have identical first and second order moments, the shortcomings of moment-based model discrimination are becoming increasingly clear. Does the situation improve if we compare whole count distributions? To establish that the Gamma-OU and CIR models are in principle discriminable, we performed an in silico experiment: first, (i) we generated noise-free synthetic data from

the CIR model for many different parameter sets; then (ii) we compared the goodness-of-fit of each model to this synthetic data.

We chose to quantify the relative goodness-of-fit of each model via the Bayes factor, which is the ratio of the likelihood of each model given the data. Concretely, we computed the log Bayes factor ... for each of the synthetic data sets we considered. A log Bayes factor of close to zero means that neither model is preferred, while a log Bayes factor of magnitude at least 2—i.e., one model is at least a hundred times more likely than the other—is commonly considered decisive evidence that one model is superior, and will be used as our criterion for distinguishability.

Because we expect that model distinguishability primarily depends on distribution 'shape' (e.g., Poisson-like or negative-binomial-like), and because shape appears to be controlled by where a parameter set resides in the two-dimensional space depicted in Figure 2, we chose 100 parameter sets that uniformly cover this space. Parameters which do not strongly control distribution shape (the average transcription rate, splicing rate, and degradation rate) were held fixed."

Certain technical details of the implementation were moved to the Methods section or SI.

- The Discussion section is written more carefully and clearly and in my opinion the authors should use a similar approach on how they present their results in order to restructure and strengthen some paragraphs in the Results subsections.
- The authors propose a method for doing Gillespie simulations with time varying parameters. There are several algorithms in the literature for such tasks (for example see Shahrezaei et al Mol Sys Biol 2008 and Voliotis et al PLoS Comp Biol 2016). How does their proposed approach related to the previous methods.

The approach by Shahrezaei et al. is approximate: formally, it appears that instead of identifying the exact τ that produces the identity in Equation 1, the authors use a left Riemann sum ($a(t)$ as a series of step functions) or a trapezoidal rule ($a(t)$ as a piecewise linear function) approximation to the integral. The authors find that "it may be difficult to analytically find t for a complex $a(t)$ ". The approach by Voliotis et al. is rejection-based, which creates the risk of unduly increasing simulation time, and uses an approximate (apparently linear) interpolant to estimate values off the grid.

We find that this task is challenging in general, and use a Riemann sum approximation to approximate the corresponding integral for the CIR model. We cite the Shahrezaei and Voliotis algorithms in the revised version of the section, and clarify these connections:

"Analogous schema have previously been used in the simulation of stochastic biological systems; the τ calculation procedure is equivalent to approximating the true reaction flux by a piecewise constant function on a finely spaced grid."

However, we find that it *can* be analytically treated for the Γ -OU model, as well as Ornstein-Uhlenbeck processes driven by compound Poisson processes in general. This is useful for computational tractability whenever a very fine time grid would be necessary to fully capture the details of the transcription process. For example, in the case of the bursty limit with short-lived spikes of activity, the Riemann sum and trapezoidal rule integral approximations require grids with vanishingly small spacing, and the rejection procedure becomes highly inefficient. The characterization of this analytical case is the key innovation of the simulation procedure.

Reviewer #2 (Remarks to the Author):

Summary:

In their manuscript, Gorin et al. discuss mathematical models based on stochastic differential equations and chemical master equations (SDE-CME) addressing transcription rate variation. The authors argue that the analysis of models that are tractable and interpretable can help in identifying the most discriminative data between competing model hypotheses and, hence, in gaining mechanistic understanding of the molecular processes underlying available transcriptomic data.

In accordance with empirical observation, RNA count distributions are often assumed to be negative binomially distributed. The authors point out that this assumption lacks direct interpretability and mechanistic insight with respect to the source of single-cell transcription rate variation. Hence, they analyze two alternative mathematical models, both tractable and interpretable: the Gamma Ornstein-Uhlenbeck (Γ -OU) model, assuming DNA stress and stress-relieving topoisomerases to underlie transcription rate variation, and the Cox-Ingersoll-Ross (CIR) model, assuming local density fluctuations of regulatory molecules responsible for transcription rate variation. Mathematical analysis reveals that both models are governed by different probability distributions but exhibit the same summary statistics. Using a simulation study, the authors show that (1) multimodal data (meaning nascent + mature RNA counts) enables better discrimination between the two models, (2) the Γ -OU and CIR models are well distinguishable from simpler models (Poisson and static mixture models), and (3) the nascent and mature RNA count distributions of both models for maximally divergent parameter sets are still visually similar. Finally, Gorin et al. show that the model parameters κ and θ of the Γ -OU model can only be reliably inferred for strongly non-Poisson cases.

The manuscript is nicely structured, well written, provides detailed methods, and is easy to follow.

We acknowledge that the mechanistic models for gene expression discussed and solved in the manuscript are definitely noteworthy, and that model inference on simulated data is interesting. However, to have a considerable impact on the field, we think that the proof-of-concept application to experimental data is mandatory to show the practical scope of the method. This is unfortunately missing in the current manuscript.

* Major issues:

Significance:

- For broad impact, the authors should showcase how their approach can be applied to experimental data to discriminate between the two proposed models (Γ -OU and CIR).

We have substantially revised the Results section of the manuscript, applying the procedure described in Figure 3 to identify the models that most effectively recapitulate scRNA-seq data. We perform this comparison for four biological replicates, with data from a single mouse neuron cell type that appears to concord with our biological assumptions of homogeneity and stationarity.

In our analysis, we find that it is possible to identify genes as better described by one or the other model with a high level of statistical confidence. To facilitate exploratory analysis on a genome-wide scale, we implemented an *ad hoc* filtering procedure. From Figure 3a, we know that the models are best distinguishable in the high-gain regime. Therefore, to identify genes which are likely to be Γ -OU-like, CIR-like, or mixture-like (not easily distinguishable), we fit the three reduced models that emerge in the high gain and slow mean-reversion limits -- i.e., the bursty model, the quasi-bursty model, and the mixture model. Maximum likelihood estimates (MLEs) of their parameters can be obtained by gradient descent in three dimensions. By evaluating Akaike information criteria at the MLEs, we select genes which were consistently assigned to a particular limit for five subsampled cell populations in a single mouse sample. The inference procedures are described in Section S8, and are implemented as native functions of the *Monod* package.

Next, we fit the full models using gradient descent in four dimensions on out-of-sample datasets and found that the qualitative predictions were largely recapitulated, with putative Γ -OU-like genes typically having likelihood ratios in favor of the Γ -OU model, putative CIR-like genes having likelihood ratios in favor of the CIR model, and putative mixture-like genes having likelihood ratios near unity and not consistently favoring either of the models. Therefore, although the *ad hoc* filter based on limiting regimes is not perfect, it is qualitatively consistent and computationally facile, providing a useful starting point for analysis. Further, we found that the models favored by the likelihood ratios tended to be consistent between four biological replicates, suggesting that the differences in model assignments correspond to biological mechanisms shared between organisms.

In a subset of the fits, the gradient descent procedure failed to achieve an acceptable fit to data at the maximum likelihood estimate (MLE). We discarded these cases. To ensure the likelihood ratios identified by the gradient descent-based procedure reflect real differences rather than incidental failure to converge, and do not grossly overestimate confidence in a particular model, we fit a subset of twelve genes using Monte Carlo Markov Chain sampling of the parameter posterior and computed Bayes factors from these fits' marginal likelihoods. The results were quantitatively consistent with the MLE-based procedure, suggesting that, with the current workflow and gene selection, the thresholds are sufficiently strict to compensate for potential errors in optimization, and the contribution of uncertainty beyond the MLE is minimal. The results of the procedure are displayed in Figure 4, which illustrates the selection procedure in panel a, visualizes the out-of-sample likelihood ratios and Bayes factors in panel b, demonstrates a set of illustrative successful fits in panel c, and displays the parameter estimates in panel d.

In sum, our analysis provides a route to identify model differences in real data by implementing and evaluating the following key points:

1. Genes that better match one or the other model can be efficiently identified by comparing fits of the corresponding reduced models, suggesting that such investigations can scale to transcriptome-wide data.
2. When compared *via* likelihood ratios computed from the full models, biological replicates yield consistent results, suggesting promising targets for investigation in future experimental studies.
3. The likelihood ratios are quantitatively consistent with the Bayes factors, suggesting that, as long as sources of numerical error are carefully controlled, the point estimate-based procedures produce results representative of more computationally intensive and statistically rigorous Bayesian analyses.

- If the approach cannot be applied to existing experimental data, the authors should elaborate on how a suitable experiment should be designed.
- While the authors clearly state in lines 42-43: “While we will not actually implement the entire closed loop paradigm depicted in Figure 1a, our work constructs one possible mathematical and computational foundation for it.”, we would have loved to see the example presented by Gorin et al. to be taken further, e.g., by showing that experimentally measured nascent and mature RNA count distributions can be better described by either the Γ -OU or CIR model in comparison to the standard negative binomial distribution assumption.

We perform this analysis in the Results section and discuss the procedure in full detail in Section S8. Figure 4 illustrates cases where the data are more concordant with one of the two SDE-based models, and contrasts them with a case where the models are not mutually distinguishable based on the available data.

Novelty:

- In the abstract the authors write “We argue that answering this question requires quantitative models that are both interpretable (describing concrete biophysical phenomena) and tractable (amenable to mathematical analysis)”. The idea of using mathematical analysis and simulations to identify whether one can discriminate between competing model hypotheses however is not novel, and a pillar of systems biology.

We fully agree with this response, and the abstract is intended to affirm it: the use of mechanistic modeling has been an essential and ubiquitous part of transcriptomics for some thirty years, and a founding principle of mathematical microbiology in general. We hope to affirm it as a guiding principle for the subfield of single-cell RNA sequencing, which typically uses descriptive rather than mechanistic analysis methods. We have previously discussed the historical dichotomy between these approaches (Gorin et al., *bioRxiv*, 10.1101/2022.02.12.480214v1, 2022, p. 6), and a key goal of the current article is to motivate the use of mechanistic models. To this end, we have used the current framework to analyze single-cell RNA sequencing data. Further, we have edited the abstract to emphasize that mechanistic modeling is not contraposed relative to the field of systems biology, and the biophysical approach and findings are not solely restricted to scRNA-seq:

Answering this question requires quantitative models that are both interpretable (describing concrete biophysical phenomena) and tractable (amenable to mathematical analysis)...

- In a recent preprint (Direct simulation of a stochastically driven multi-step birth-death process; <https://doi.org/10.1101/2021.01.20.427480>) Gorin and Pachter present a detailed discussion of the Γ -OU model and a simulation method for the corresponding SDE–CME. This limits the novelty of the present manuscript. Also, an explicit reference to this work of the same authors is missing.

We apologize for the lack of clarity on this point. We omit references to this preprint for the same reason we omit references to <https://www.biorxiv.org/content/10.1101/2021.09.06.459173v4>: although their scopes differ, both are integral parts of the current report, and we seek peer review on all of its components. We reproduce the key technical parts of the simulation-focused preprint, in condensed form and with several edits for clarity, in the supplemental section “Simulation of the Gamma Ornstein-Uhlenbeck model”; in particular, we revise the notation to reduce confusion between the arrival times and the reaction time step, both denoted by τ . The exploratory analysis performed in the preprint (e.g., the demonstration of limiting cases in Figure 3) has been superseded by the comprehensive analysis in the current article.

- In the abstract the authors write: “We introduce a framework to numerically generate these predictions and analytically unify known transcriptional regimes.” Is the simulation method that the authors refer to the same as the one that has been put forward in the preprint?

The method for exactly simulating the Γ -OU model, one of the constituent parts of the current article, was originally reported in the simulation-focused preprint. The method for approximately simulating the CIR model has not been previously reported, but uses an approach similar to existing studies.

- The authors should more strongly elucidate the novelty of current study with respect to Ref. 80 and emphasize its significance to the field.

[Redacted]

[Redacted]

* Minor issues:

Wording/Text:

- Please define the term 'mean reversion' in equation (4) similar to its description in caption Figure 1b.

The paragraph now reads:

"...for some coefficients A and B, where [mean reversion] denotes a term that drives the process toward a deterministic level and [noise] denotes a model-dependent term that introduces stochastic variation..."

- In line 125: The acronym CLE is not defined.

The acronym is now defined earlier in the same section.

Presentation:

- The upper points in Figure 1a “Hypothesis-driven transcriptomics experiment” and “model rejection be incorporated” are not addressed in the manuscript, which is touched on in the main text but might not become apparent from cross-reading the paper.

The following sentence has been added to the introduction, so that the main text better reflects the idea of Figure 1:

“One can then perform this experiment, use the results to refine or reject those models, and iterate.”

- Figure 1c: We suggest to highlight that the dots in the schematic describing the Γ -OU model represent the stress-relieving topoisomerases, while the same dots in the schematic describing the CIR model represent regulatory molecules.

We have substantially revised the figure to illustrate the modeled physical processes in more explicit detail.

- Figure 1c: What do the colors represent? E.g. orange stress arrow?, pink dot?, green relief arrow?, orange DNA coil? If important, please add a legend or labels to the figure. Same holds for the arrows in Figure 1b.

We have revised the conceptual illustrations in Figure 1 to clarify the chemical reactions involved in the candidate models.

- Eq. 1: Please mention why decay of nascent RNA is not considered here.

We omit the discussion of nascent RNA decay based on standard conventions for modeling eukaryotic cells (cf. Cao and Grima, *PNAS*, 117:9, 4682-4692, 2020), as well as physical considerations: to a first-order approximation, RNase is physically excluded from the nucleus by the nuclear membrane. However, if a gene of interest does undergo nuclear RNA degradation, it is relatively straightforward to self-consistently model this reaction pathway. We discuss the mathematics of this approach, and generalizations to more complex downstream processes, in Section S9:

“The Γ -OU and CIR solutions extend to arbitrary directed acyclic graphs of splicing and degradation with no loss of generality. Specifically, our typical reaction system involves the following dynamics downstream of the promoter... This system corresponds to the following PDE characteristic: ... More general downstream processes can be incorporated by defining a more general U1 according to Equations 12 and 13 of [the Biophysical Journal article]. For example, if the degradation of nascent transcripts competes with splicing -- i.e., the system includes the [nascent degradation reaction] -- the following characteristic should be used to solve the system...”

- Line 83: Has ‘constitutive model’ been defined?

We have added a reference to Equation 1, the reaction network that defines the constitutive model.

- Eq. 6: Please make $\varepsilon(t)$ more explicit by emphasizing that a and θ enter here.

To clarify that $\varepsilon(t)$ depends on both a and θ , we have switched to the notation $\varepsilon(t; a, \theta)$ in the main text.

- Figure 2b: Are the plots shown schematics or results of simulations? Please specify.

We agree that the original version of the figure, which featured a mix of schematics and simulated data, was confusing. To alleviate this confusion, we have reimplemented the entire figure, using only simulations. We also make this more clear in the associated text and figure caption.

In particular, for each of the four depicted limiting models, we performed 8 transcription rate simulations. These are depicted in the $K(t)$ vs time column. For two fairly arbitrary values of t_1 and t_2 (which are about 20 minutes apart according to the figure timescale), we take the values from that column and visualize them in cell cartoons. Because the transcription rate can vary over many orders of magnitude in the bursty regimes, we chose the darkness of the cell colors to depend on K roughly as $\exp(-K/\langle K \rangle)$.

These cartoons are all generated in Python immediately after the simulations. Separately, the plots in the $P_{ss}(x_N, x_M)$ are generated from sampling the relevant steady state probability distributions.

The notebook that generates all of these plots is made available on the project’s GitHub page, and is in Colab for ease of access.

- Figure 2b: Please state somewhere in the figure caption that the green shades of schematics $P(K, t_1)$ and $P(K, t_2)$ represent the values of K .

This clarification has been added to the figure caption:

“...transcription rates are visualized using trajectories and cell cartoons, where transcription rate is a logarithmic function of cell color.”

- Figure 2b: The schematics of $P(K, t_1)$ and $P(K, t_2)$ for Γ -OU and CIR look bimodal, which does not fit the respective schematics for $K(t)$. Please adapt the green colors in the schematics $P(K, t_1)$ and $P(K, t_2)$

to show a more representative image, maybe similar to your tweetorial 7/
<https://twitter.com/lpachter/status/1436344431657979913>.

By reimplementing Figure 2, we have bypassed the problem of potentially arbitrary or misleading schematics. Now, the cell cartoons are all generated in Python from the data obtained by transcription rate simulation. They turn out to look somewhat bimodal, for the reason that (in the bursty regimes), at a given time, some cells are highly active and some cells are not very active.

- Figure 2b: The schematics of $P_{ss}(x_N)$ of the mixture model and Γ -OU look identical. However, they should have different probability distributions for nascent RNA counts. Please adapt if only schematics and no simulations.

To partly correct this, we have chosen to depict joint steady state probability distributions instead of the nascent marginal distributions.

Nonetheless, this is actually an interesting point about the underlying biology. Under particular parameter choices, both models converge to the same distribution shape for the nascent mRNA distribution: the high-gain or “bursty” Γ -OU limit induces a negative binomial law for the nascent counts that cannot be distinguished from the negative binomial law for the “slow” mixture model limit. This has previously been noted as a challenge to model identification (Ham et al., *Phys. Rev. Lett.* 124, 108101, 2020). The referenced investigation partially motivated the current study, as well as a dedicated discussion of these limits (Gorin and Pachter, *bioRxiv*, 10.1101/2020.09.25.312868, 2020), which found that the transcriptional hypotheses *are* distinguishable in a two-species model.

- In lines 143 and 144 you state that: ‘However, the CIR model predicts a novel family of count distributions with heavier tails than their Γ -OU counterparts.’ This is not easily visible in Figure 2b. We suggest to graphically highlight the differences between the $P_{ss}(x_N)$ schematics for Γ -OU and CIR more strongly.

To address this, we switched to depicting joint distributions, for which the differences in tails are more visible (although they are still subtle). One can see from the rightmost column of Figure 2 that there are more ‘loose dots’ around the quasi-bursty 2D histogram than the geometric burst histogram.

The differences between distributions are now also specifically emphasized in the supplementary Figure S4.

- You only show the probability distribution of nascent RNA counts in Figure 2b. As you also identify differences between the processes in the mature RNA compartment, one could already allude to this in Figure 2b as an extra column $P_{ss}(x_M)$.

We have revised and simplified this illustration by demonstrating the entire bivariate distributions.

- Figure 3a: How different do the results look for a different set of ground truth parameters $\langle K \rangle$, β , and γ as defined in line 189? Given that the interpretability of the models plays a crucial role, is there a specific reason to focus on the ground truth parameters defined in line 189?

We hope that the inference on real data we perform in the updated Results section partly addresses this question: given biological count data with an ostensibly unknown underlying model and parameters, we can fit parameters for a wide variety of distributions and control for model misspecification.

We have also added Figure S6, which reproduces Figure 3a under slightly different assumptions (if the data is sampled from the GOU model instead of the CIR model; if the average transcription rate is higher; and if beta is larger than gamma). In all circumstances, one obtains extremely similar results.

- Figure 3a: Please show the color legend as a gradient scale with corresponding $\log_{10}(\text{BF})$ values.

Figure 3c now has the quantitative colorbar demonstrating the $\log_{10}(\text{BF})$ values.

- Please add the axis title $\log_{10}(\text{BF})$ in Figure 3c for clarity.

Figure 3c now names the ordinate in the title.

- In lines 205 and 206 the authors state that ‘... and found that even when predictions are maximally divergent, the distributions are still visually alike (Figure 3d).’ Is a statistical test able to reject the hypothesis that the probability distributions of the nascent and mature RNA counts for Γ -OU and CIR are from the same underlying distribution?

We have revised this panel. The panel still illustrates that visible qualitative differences between the models at a given set of SDE parameters are fairly minor and subtle. However, the specific set of parameters is now a maximum-divergence point from Figure 3a. This revision accomplishes two goals. Firstly, it immediately answers the statistical question: Figure 3a demonstrates that the Bayes factor *does* exceed a typical criterion for model selection whenever the dataset contains 200 or more cells. Secondly, it bypasses some of the limitations of the gradient descent procedure used to obtain the original maximally divergent parameter set, including limited reproducibility due to the procedure’s computational intensity, relevance to finite datasets, and typical problems associated with optimization of potentially non-convex functions, such as numerical errors in gradient evaluation and the lack of guarantees that the discovered optimum is global.

- Figure 3e: Do these results also hold for the CIR model?

We hope that the analysis in the revised Results section, which compares the models using likelihood ratios and Bayes factors, addresses this question. The parameter recovery results are displayed in Figure 4d, and the raw data for the Bayes factors fits are available in the associated GitHub repository.

- Figure 3e: The middle plots showing the nascent versus the mature RNA counts for an overdispersed and Poisson-like case are distracting. If the authors find it important to visually exemplify this, these plots could be moved to Figure 2b, where the models and their probability distributions are introduced.

We have cut the superfluous histograms from Figure 3, and took the reviewer's suggestion, putting similar plots into Figure 2.

- Figure 3e: Please add a colorbar.

Figure 3c now has the quantitative colorbar demonstrating the probability densities.

- In lines 211 and 212 it is stated that "To simplify this computation, we assumed that β , γ , and $\langle k \rangle$ were known, because they can be accurately and robustly inferred from empirical means (one can also imagine performing separate experiments to determine them first)". How would the results from Figure 3e look like if the parameters β , γ , and $\langle k \rangle$ were indeed obtained from simulated data?

We choose this assumption by appealing to the central limit theorem. Assuming $\langle k \rangle = 1$, the parameters $1/\beta$ and $1/\gamma$ are the expectations of the nascent and mature RNA counts. These expectations can be estimated in an unbiased way by setting them to the empirical means, with resulting uncertainty (square root of the mean squared error) of σ/\sqrt{n} for n cells. Given $n > 3300$ in the investigated datasets, this is typically sufficient to make the error negligible.

Nevertheless, the distribution of β does not necessarily have the same error behavior as the distribution of $1/\beta$, and it is appropriate to fit all four parameters simultaneously. The results of this analysis are shown in Figure 4, performed using gradient descent for 69 genes (with 57 yielding usable results) and MCMC for a subset of 12 genes. The details of both inference procedures are described in Section S8. We hope that this analysis, which does not make assumptions regarding the distribution of the parameter posteriors, addresses this question.

REVIEWERS' COMMENTS

Reviewer #1

The authors have fully addressed my comments.

Reviewer #2

The authors addressed all our points raised. Overall, it's great that the method is now applied on real data. However, the revised manuscript also raises a couple of question with respect to this analysis, and I am missing a discussion of the validity and sensibility of the findings. Finally, the fitting procedure sounds relatively complex. It would be great if authors could comment on how easy it is for a computational researcher to reproduce the analysis shown in Fig. 4b

A few specific issues:

- Please detail the stringent filtering procedure in the main part of the manuscript. Can you be sure that it does not bias your results? What do you get if your filter is less stringent?
- Can you motivate the p value of 10^{-8} ? Did you correct for multiple testing?
- Fig 4a: Please explain when you assign the mixture-like mode. What's the difference between mixture-like and gray?
- Fig 4b: Why do some genes have boxplots, others not? Why do some have black circles? Why is there only one data point for so many genes? I miss the data points that belong to Gabra4
- Fig 4d: What is the dot size supposed to convey?
- Methods / Filtering: Why sampling 3k from 3787 genes?
- Can you discuss why the genes end up in the one or the other cluster? Is there evidence that a gene might be more prone to DNA regulation vs TF regulation?

Reviewer responses

Reviewer #1 (Remarks to the Author):

The authors have fully addressed my comments.

Reviewer #2 (Remarks to the Author):

The authors addressed all our points raised. Overall, it's great that the method is now applied on real data. However, the revised manuscript also raises a couple of question with respect to this analysis, and I am missing a discussion of the validity and sensibility of the findings.

We have added a brief discussion of the results under “Multimodal count distributions in sequencing datasets suggest distinct modes of transcriptional regulation,” summarizing the findings and caveats.

The likelihood ratio procedure yields results that are (i) similar to the distribution shapes observed in the raw data, up to possible numerical errors, (ii) broadly consistent with the predictions from the reduced model fit, although some discrepancies do occur, particularly for 'mixture-like' genes that exhibit higher identifiability under the full model, (iii) qualitatively consistent between datasets, and (iv) largely, but not perfectly, coherent with a full Bayesian procedure. Therefore, the distributions associated with the proposed models can be distinguished in practice. Further, these differences can be probed using a range of tools, some more approximate and suited to genome-wide exploratory analysis, others more statistically rigorous and suited to detailed study of gene targets.

We hope that the modifications to the filtering and fitting procedures, the expanded documentation in the manuscript, and the responses to the questions below clarify the statistical validity and sensibility of the procedure. As we focus on the theoretical and numerical questions, i.e., whether the models can be distinguished and whether these model distinctions are reproducible, we stop short of discussing the implications of the findings in biological terms.

Finally, the fitting procedure sounds relatively complex. It would be great if authors could comment on how easy it is for a computational researcher to reproduce the analysis shown in Fig. 4b

We anticipate this to be relatively straightforward, albeit computationally intensive. We have added the following statement under “Code Availability”:

The GitHub and Zenodo repositories include scripts used to construct a mouse genome reference, pseudoalign datasets, and generate all figures. They are modular: the analysis can be restarted at a set of intermediate steps. The outputs of certain steps, viz. pseudoaligned count matrices, results of the Monod pipeline, the list of genes of interest, results of the gradient descent

procedure, and results of the Bayes factor computation procedure can be recomputed, or loaded in based on files available in the repositories.

We report the versions of the software used to run the analysis in the same section.

To simplify this procedure further and make it more transparent, we have removed the dependence on pre-extracted count matrices; analyses now begin from the full dataset output by *kallisto|bustools*. In the process of this revision, we corrected an error whereby a set of the barcodes corresponding to each cell type assignment were inadvertently omitted. The current version of the analysis now uses the full dataset for each cell population.

Finally, we give instructions, as well as approximate timings, for reproducing all parts of the analysis from intermediate points in the GitHub repository README document.

A few specific issues:

- Please detail the stringent filtering procedure in the main part of the manuscript. Can you be sure that it does not bias your results? What do you get if your filter is less stringent?

We have revised the filter and added further characterization in the Methods section, describing the steps as well as the motivation for performing them:

Within Monod, the SciPy implementation of L-BFGS-B was used to perform gradient descent and obtain maximum likelihood estimates for the three-parameter reduced models. We selected genes most consistently assigned to each model (Figure 4a) according to their Akaike weights. This step identified genes that appeared to be reproducibly described by each model class, and provided a tentative basis for out-of-sample predictions. Finally, we restricted our analysis to the best-fit 35 genes in each category, as quantified by the maximum rank of the chi-squared statistic observed across the five subtype datasets. This filtering step was applied to avoid contributions due to model misspecification or poor convergence, and focus on the genes that best agreed with the regimes of interest. The preliminary analysis produced 35 genes of interest for the Gamma-OU-like and CIR-like categories and 10 genes for the mixture-like category.

If the filter is less stringent, we obtain more ambivalent results, with a large number of genes that are not consistently assigned to one or the other model, or suggest failure to converge. As we are principally interested in considering the models motivated by the theory, we seek to illustrate the least ambiguous cases, to motivate the models' use in biophysical analyses.

This approach, which enriches for the most promising genes, is naturally biased toward the extreme cases. We anticipate that more comprehensive treatment of the more typical cases observed across the transcriptome is a useful future research direction. However, pursuing it would require another round of substantial improvements to the numerical implementation of these models.

- Can you motivate the p value of 10^{-8} ? Did you correct for multiple testing?

This p -value threshold was imposed as an *ad hoc* filter for goodness of fit. The magnitude was chosen in light of the large number of replicates under the model (approximately 5000 cells per mouse); choosing a higher threshold would lead to the rejection of fits which appeared to be consistent with the data on visual examination. We did not correct for multiple testing, as in the current exploratory context, we considered it more appropriate to err on the side of overrejecting than keeping in genes with results that failed to converge to a satisfactory fit.

In the current version, we show all of the log-likelihood ratios (within the domain $[-150, 150]$) in lieu of imposing potentially arbitrary goodness-of-fit thresholds late in the analysis. Further, we display the raw data to clarify whether retaining the data is appropriate.

- Fig 4a: Please explain when you assign the mixture-like mode. What's the difference between mixture-like and gray?

The mixture-like mode corresponds to the genes that are consistently better described by the bivariate negative binomial distribution (Equation 7 in Supplementary Note) than either the Gamma-OU limit (Equation 8 in Supplementary Note) or the CIR limit (Equation 9 in Supplementary Note). The gray dots correspond to the genes that are not consistently assigned to a particular model in the subtype datasets; for example, their data may be too sparse to identify the models, so the different cell subtypes give contradictory results. Fig 4a only displays a single subtype, whereas the colors are based on thresholding the results in all subtypes. We omit the inconsistent cases, but we consider the mixture-like cases further, as they may reflect the slow-driving limit of the two models.

- Fig 4b: Why do some genes have boxplots, others not? Why do some have black circles? Why is there only one data point for so many genes? I miss the data points that belong to Gabra4

The boxplots and black circles denoted the distributions of Bayes factors computed for those genes. In the current revision, we have replaced the box plots with horizontal markers, which directly show the underlying data points. These markers are connected by a vertical line, which guides the eye and demonstrates the range of observed Bayes factors. We do not compute Bayes factors for all of the genes due to the considerable computational intensity of this procedure.

In the previous revision, we excluded results with potentially poor convergence (according to a chi-squared p -value threshold) as well as apparently excessively high log-likelihood ratios (according to the plot bounds of approximately $[-125, 125]$). In the current revision, we have somewhat simplified this procedure. Now, we only perform the second step, with the wider bounds $[-150, 150]$. We show all obtained likelihood ratios for all genes, as long as they are within these bounds, without performing further goodness-of-fit filtering.

The choice to display all of the data is partially motivated by the high rejection rate observed in the previous version. We anticipate that this stems from fundamental statistical considerations: given thousands of cells, which are treated as independent replicates, the chi-squared procedure will reject often, with high confidence (e.g., Lin, Lucas, and Shmueli, *Information Systems Research*, 2013, 10.1287/isre.2013.0480). A more informative procedure should combine chi-squared p -values with estimates of effect size. However, we have only seen a limited amount of previous research on using such thresholds for goodness-of-fit, and strive to motivate this problem as a direction for future statistical investigation.

To more directly clarify whether keeping the data is justified, we have added Supplementary Figures 8-37. These figures plot the raw spliced and unspliced data, as well as the best model fits, for each dataset and gene. Although the marginal distributions only show a part of the full picture, we find that the arbitrary threshold of [-150,150] performs reasonably. Genes with higher-magnitude log-likelihood ratios tend to have fits very different from the data, suggesting that they failed to converge. On the other hand, genes with lower-magnitude log-likelihood ratio tend to have fits that broadly recapitulate the shape of the data.

- Fig 4d: What is the dot size supposed to convey?

The large dots in Fig 4d denote the (named) genes illustrated in Fig 4c. We clarify this point in the revised caption:

(colors: predicted regimes, large dots: genes illustrated in panel c. Genes with absolute log-likelihood ratios above 150 have been excluded)

- Methods / Filtering: Why sampling 3k from 3787 genes?

We typically use this approach in the *Monod* pipeline to achieve consistency among datasets of varying quality: different datasets typically have different sets of genes that meet the filtering thresholds. This approach ensures that a predefined number of genes undergoes analysis. If the data for some genes are too sparse to draw statistical conclusions, they can be filtered *post hoc*.

However, in the current revision, we remove this source of potential variation, and fit all 3,677 genes that pass the filtering step in all four datasets.

- Can you discuss why the genes end up in the one or the other cluster? Is there evidence that a gene might be more prone to DNA regulation vs TF regulation?

We did not observe consistent trends with respect to specific regulatory motifs, as typical databases (e.g., Gene Ontology or enrichR) characterize associations with particular pathways, rather than classes of mechanisms. In the current work, we restrict ourselves to noting that

distribution shapes associated with particular mechanisms persist in different samples, and may reflect some underlying difference in regulatory biochemistry, but the limitations of the models cause us to stop short of speculating about particular genes. We anticipate that the current suite of mathematical and analytical tools will be helpful in asking and answering such questions, especially combined with genome-wide data reporting some proxy for the gene state (e.g., various experimental methods with ChIP-seq and ATAC-seq combined with RNA-seq). However, obtaining biological insights will require careful benchmarking and calibration with reference to known regulation pathways, particularly using sets of genes that are co-expressed due to proximity on a chromosome or co-regulation by a common TF.